# SpiDe-Sr: blind super-resolution network for precise cell segmentation and clustering in spatial proteomics imaging

Rui Chen[1,2,3], Jiasu Xu[1,2,3], Boqian Wang[1,2], Yi Ding[1,2], Aynur Abdulla[1], Yiyang Li [2], Lai Jiang[1] & Xianting Ding [1,2] ✉

Spatial proteomics elucidates cellular biochemical changes with unprecedented topological level. Imaging mass cytometry (IMC) is a high-dimensional single-cell resolution platform for targeted spatial proteomics. However, the precision of subsequent clinical analysis is constrained by imaging noise and resolution. Here, we propose SpiDe-Sr, a super-resolution network embedded with a denoising module for IMC spatial resolution enhancement. SpiDe-Sr effectively resists noise and improves resolution by 4 times. We demonstrate SpiDe-Sr respectively with cells, mouse and human tissues, resulting 18.95%/27.27%/21.16% increase in peak signal-to-noise ratio and 15.95%/31.63%/15.52% increase in cell extraction accuracy. We further apply SpiDe-Sr to study the tumor microenvironment of a 20-patient clinical breast cancer cohort with 269,556 single cells, and discover the invasion of Gram-negative bacteria is positively correlated with carcinogenesis markers and negatively correlated with immunological markers. Additionally, SpiDe-Sr is also compatible with fluorescence microscopy imaging, suggesting SpiDe-Sr an alternative tool for microscopy image super-resolution.

Spatial proteomics could elucidate tumor microenvironment[1,2], organ heterogeneity[3], and cellular biochemical changes occurring at different stages of disease[4–6]. Imaging mass cytometry (IMC) is a targeted spatial proteomic technique that avoids ripping cells out of their native environments by coupling immunocytochemical methods with laser ablation[7,8]. To achieve accurate cell extraction and cell clustering for subsequent statistical analysis, IMC imaging with high peak signal-to-noise ratio (PSNR) and rationalized details is desired. However, due to the non-specific binding of antibodies, IMC imaging is susceptible to noise contamination, especially in the case of multi-channel staining[2,8]. Meanwhile, IMC imaging resolution is limited by the size of laser spot, as each pixel in the image process is generated by laser ablation of the metal-labeled tissue[8].

The most straightforward strategy to obtain IMC images with high PSNR is to manually eliminate pixel values above and below empirical thresholds, but relatively ultra-high and ultra-low expression of markers would also be eliminated together[9]. More effective strategies include selecting specific antibodies that rarely cross react with non-target antigens[10]. As for improving IMC imaging resolution, the most fundamental strategy is to reduce the laser spot size. There is no comprehensive solution so far because the laser spot size is a complex parameter that relies on the laser energy required for ablation, beam radius, and single laser ablation shot duration[8,11,12]. Enlarging the tissue with expansive hydrogel is a feasible approach for increasing image resolution. However, IMC requires dehydrating the tissue. To keep the tissue from shrinking after dehydration remains an unsolved challenge[13,14].

Apart from physical or biological methods, data-driven approaches offer an alternative opportunity to recover authentic information from noise contaminated images and reduce human labor. The

[1]Department of Anesthesiology and Surgical Intensive Care Unit, Xinhua Hospital, School of Medicine and School of Biomedical Engineering, Shanghai Jiao Tong University, Shanghai, China. [2]State Key Laboratory of Systems Medicine for Cancer, Institute for Personalized Medicine, Shanghai Jiao Tong University, Shanghai, China. [3]These authors contributed equally: Rui Chen, Jiasu Xu. ✉e-mail: dingxianting@sjtu.edu.cn

classical approach is to train the supervised deep learning network that learns the mappings between image pairs of low PSNR/resolution images and ground truth (that is, images without noise contamination or high-resolution images with the same underlying scene as the low-resolution images), which, respectively, refers to images denoising and super-resolution (SR)[15–17]. Such methods have been broadly adopted to enhance the performance of optical imaging[18–20]. However, the laser ablation during IMC imaging process precludes the same sample being acquired twice. This makes the acquisition of clean-noisy or high resolution (HR)-low resolution (LR) IMC image pairs almost impractical. With no proper ground truth to supervise the network training, conventional supervised learning methods are not applicable to IMC images. Meanwhile, unsupervised learning has also evolved rapidly in natural image enhancement (denoising or SR[21,22]). For natural images, visual quality is the priority. But for IMC images, rational enhancement is the necessary foundation for subsequent analysis. The image enhanced by the existing unsupervised SR network lacks rationality due to the absence of ground truth[20,23].

Here, we propose SpiDe-Sr (spatial proteomic images denoising and super-resolution), a blind (without true blur kernel) super-resolution network embedded with self-supervised denoising module for enhancing PSNR and cell extraction accuracy in IMC. SpiDe-Sr consisted of a self-supervised denoising module and a blind super-resolution module. The denoising module was based on the insight that pairs of noisy images generated by neighbor sub-sampling from the single noisy images could be used for training, because the noisy image pairs were conditionally independent when the gap between the underlying ground truth images of the noisy image pairs was small[24,25]. Thus, a U-net network[26] was trained with image pairs for denoising. The strategy of the blind super-resolution module was to iteratively correct the predicted blur kernel to approximate the true blur kernel, so that the image details could be rationally enhanced without additional reference[27]. The SR module was comprised of individually trained predictor, corrector, and SR network[28]. We verified the SpiDe-Sr with metal/fluorescence dual-labeled samples of MCF-7 cell line, mouse fatty liver tissue, and human breast cancer tissue, respectively. SpiDe-Sr was then applied to clinical breast cancer samples from a 20-patient cohort. The samples were stained with 14 biomarkers. With the assistance of SpiDe-Sr, we found that Gram-positive ($G^+$) and negative ($G^-$) bacteria were commonly present in the tumor microenvironments. The expression of $G^+$ bacteria marker was positively correlated with the expression of immunological markers (such as CD45), while the expression of $G^-$ bacteria marker was positively correlated with carcinogenesis markers (such as IFI6) and negatively correlated with immunological markers (such as CD68 and CD8a). In addition, we also demonstrated SpiDe-Sr was compatible with fluorescence microscopy imaging, suggesting its versatility in microscopy image processing.

## Results

### Development and performance validation of SpiDe-Sr

The general composition of SpiDe-Sr is schematized in Fig. 1a. The SpiDe-Sr comprised of two main modules, namely the self-supervised denoising module and the blind super-resolution module. In denoising module training, a pair of sub-sampled images $(g_1(y), g_2(y))$ were generated from noise image $y$ with the sub-sampler $G$. The noisy image pairs were conditionally independent when the gap between the underlying ground truth images of the noisy image pairs was small[24,25]. Therefore, $g_1(y)$ and $g_2(y)$ could be used, respectively, as the input and target to train the denoising network ($U_\theta$), which uses U-net[26] as the framework (Supplementary Fig. 1a). The loss function of $U_\theta$ consisted of two terms: the reconstruction term ($L_{rec}$) computing the differences between the output and the noisy target, and the regularization term ($L_{reg}$) computing the difference of the ground truth pixel values between the sub-sampled noisy image pair[24]. The super-resolution

module had three components trained individually in a self-supervised manner: the blur kernel predictor ($P_\theta$), the blur kernel corrector ($C_\theta$), and the image super-resolution network (SFTMD[28], $S_\theta$) (Supplementary Fig. 2a–c). The predictor took the low-resolution image ($image^{LR}$) as input and the initial blur kernel ($k_0$) as output. The initial blur kernel was iteratively corrected by the corrector to avoid super-resolution images contain artifacts due to mismatched blur kernel[27]. For the sensitivity of SR to kernel mismatch, please refer to Supplementary Fig. 3a. In each corrector iteration, a super-resolution image ($image^{SR}_n$) was generated based on the estimated blur kernel ($k_n$) until convergence (Fig. 1c). The loss of the super-resolution module ($Loss$) was calculated by the mean square error between the estimated blur kernel output from the corrector ($k_{1,2,...n}$) and the true blur kernel ($K$). After training, interpretable features and accurate super-resolution mappings were learnt by SpiDe-Sr (Supplementary Figs. 1d, 2d and 11), which could be applied to subsequent acquisitions without additional training (Fig. 1b).

To quantitatively evaluate the benchmark performance of SpiDe-Sr, the raw IMC images were served as ground truth because of the lack of clean and high-resolution images. The raw images were superimposed with noise and down-sampled to one-fourth of the original size to form blurred images. The peak signal-to-noise ratio (PSNR) and structural similarity (SSIM) were calculated between the ground truth and the blurred images before and after SpiDe-Sr enhancement (Details are provided in the Methods section). After enhancement by SpiDe-Sr, the PSNR was improved by $23.1 \pm 5.4\%$, raised from $25.25 \pm 3.28$ dB to $31.72 \pm 1.56$ dB, and the SSIM was improved by $29.17 \pm 41.70\%$, raised from $0.48 \pm 0.24$ to $0.61 \pm 0.26$ (Fig. 1d and Supplementary Fig. 1b, c). All cell segmentation tasks were implemented with the Cellpose algorithm[29]. Fewer cells were missed after enhancement by SpiDe-Sr. The accuracy of cell extraction was improved by $58.19 \pm 41.92\%$, from $60.85 \pm 13.68\%$ to $90.90 \pm 3.61\%$ (Fig. 1e, g, and Supplementary Fig. 3b–d). The improvement of the PSNR and SSIM of the images, and cell extraction accuracy were statistically significant (paired-samples two-sided t-test, $P < 0.001$). In addition, SpiDe-Sr was visually superior to three state-of-the-art single image super-resolution methods including SRCNN[19,30], KernelGAN[31], and RCAN[19,20,32] (Fig. 1f). All methods, except SpiDe-Sr, accidentally treated CD8 (red pixel points) that should not be expressed in View 2 as effective information.

### SpiDe-Sr enhanced IMC images of MCF-7 cell line

While the resolution of the raw IMC image was above 1μm as determined by the imaging principle of laser ablation[8], the resolution after super-resolution was enhanced to 250 nm, which was close to confocal images at 40× magnification (40×, 0.95NA, resolution: 330 nm). In real experiments, the resolution of the IMC image enhanced by SpiDe-Sr was faithfully close to confocal images at 20× magnification (20×, 0.4NA, resolution: 830 nm) because the tissue around the laser spot was also vaporized during IMC imaging. Therefore, we opted to use 20× confocal images as ground truth (GT) for comparison with the IMC images before and after the enhancement of SpiDe-Sr.

20× confocal and IMC images of MCF-7 cell line were acquired in pairs to quantitatively evaluate the performance of SpiDe-Sr (Fig. 2a), where Tubulin, CD45, and CD34 were stained with fluorescent/metal dual-labeled antibodies and cell nucleus were stained with both fluorescent (DAPI) and metal ($^{191}$Ir/$^{193}$Ir) dyes (Fig. 2b–d). The Tubulin, CD45, and CD34 were chosen as representatives of markers with relatively high, moderate, and low expressions, respectively, as identified in the pre-experiments. The PSNR of Tubulin, CD45, and CD34 signal intensities in the IMC images were, respectively, $25.66 \pm 1.98$, $15.39 \pm 1.30$, and $22.65 \pm 1.95$ before SpiDe-Sr enhancement, and $27.01 \pm 1.95$, $18.27 \pm 1.30$, and $24.74 \pm 1.98$ after SpiDe-Sr enhancement. Thus, SpiDe-Sr increased the PSNR by $5.36 \pm 4.06\%$, $18.95 \pm 5.89\%$, and $9.42 \pm 5.47\%$, respectively (Fig. 2e). Meanwhile, the SSIM was,

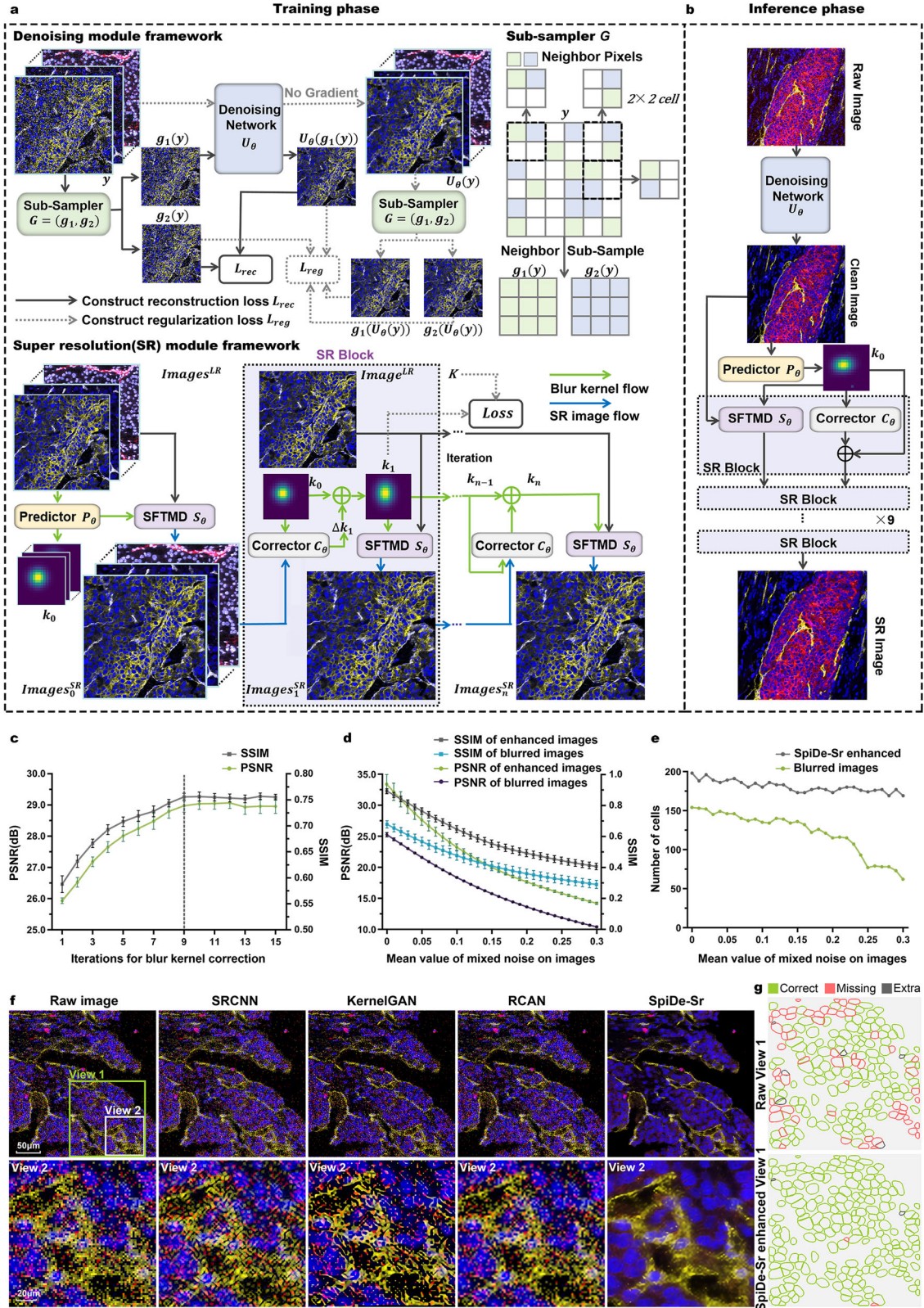

respectively, 0.76 ± 0.08, 0.52 ± 0.09, and 0.44 ± 0.14 before enhancement, and 0.88 ± 0.08, 0.7 ± 0.11, and 0.69 ± 0.11 after enhancement, which was also increased by 16.78 ± 1.81%, 40.21 ± 17.73%, and 57.01 ± 17.49%, respectively (Fig. 2f).

As the image quality was enhanced by SpiDe-Sr, the cells could be extracted more accurately (Fig. 2g, Supplementary Fig. 4a−c). The accuracy of cell extraction in the raw IMC images of Tubulin, CD45, and

CD34 being labeled was 90.03 ± 8.26%, 85.82 ± 7.65%, and 82.41 ± 9.37% prior SpiDe-Sr enhancement, while increased to 96.10 ± 4.62%, 95.86 ± 3.82%, and 94.59 ± 6.52% after SpiDe-Sr enhancement. In this study, accurate extraction of a cell was defined as being detected in both the IMC and confocal images. However, accurate cell extraction did not necessarily lead to accurate cell segmentation, which required precise determination of the cell

**Fig. 1 | SpiDe-Sr method. a** The architecture of SpiDe-Sr. The network was comprised of the denoising module and the super-resolution module. The denoising module included the neighbor sub-sampler and the U-net denoising network. And the super-resolution module had three components: the blur kernel predictor ($P_\theta$), the blur kernel corrector ($C_\theta$) and the image super-resolution network ($S_\theta$). **b** Inference using the trained SpiDe-Sr network. The architectural details and interpretability of the SpiDe-Sr were illustrated in Supplementary Fig. 1 and Supplementary Fig. 2. **c** Quantitative evaluation of SR image quality with different iterations of blur kernel estimation. Dashed line indicated the optimal number of iterations. $n = 4392$ images. **d** Quantitative evaluation of image PSNR and SSIM with different noise levels before and after the SpiDe-Sr enhancement. $n = 4392$ images.

PSNR, peak signal-to-noise ratio, larger means less noise. SSIM, structural similarity, larger means more similar to the ground truth. In (**c**, **d**), data were mean ± SD. **e** The number of cells extracted based on images with different noise levels before and after the SpiDe-Sr enhancement. Total number of cells in the field of view was 200. **f** Visual comparison of SpiDe-Sr method with the three state-of-the-art (SOTA) super-resolution methods including SRCNN, KernelGAN, and RCAN. **g** Spatial profiles of extracted cells in the field of View 1. Correctly segmented regions (true positives) were colored in green. Missing (false negatives) and extra regions (false positives) were colored in red and gray, respectively. All cell segmentation tasks in our work were implemented with the Cellpose algorithm. Source data are provided as a Source data file.

boundaries. Therefore, intersection over union score (IoU) was calculated to evaluate the accuracy of cell segmentation[21,33]. SpiDe-Sr increased the IoU for 87.04%, 96.68%, and 96.36% of the extracted cells in Tubulin, CD45, and CD34 images, respectively (Fig. 2h). The improved accuracy of cell boundary resulted in more accurate detection of protein expression levels. In the enhanced images, the marker expressions in the accurately extracted cells were closer to that in the confocal images than raw images (Fig. 2l, j).

In addition, the performance of SpiDe-Sr was comprehensively compared with three prevalent super-resolution methods, including SRCNN, KernelGAN, and RCAN on the cell line images (Fig. 2k, l, Supplementary Fig. 4d–h). The running time of SpiDe-Sr (0.45 ± 0.02 s/pic) had no advantage over RCAN (0.35 ± 0.01 s/pic), the best performing of the three methods, but SpiDe-Sr was better than RCAN with 17.78 ± 8.72% higher PSNR and 32.28 ± 15.20% higher SSIM increase after image enhancement. SpiDe-Sr was also superior to the other three methods in terms of visual representation (Fig. 2l).

### SpiDe-Sr enhanced IMC images of mouse fatty liver tissue
To quantitatively evaluate the performance of SpiDe-Sr on animal samples, paired 20× confocal and IMC images of mouse fatty liver were acquired (Fig. 3a). Tubulin, CD45, and CD34 were stained with fluorescent/metal dual-labeled antibodies and cell nucleus were stained with both fluorescent (DAPI) and metal ($^{191}$Ir/$^{193}$Ir) dyes. (Fig. 3b–d). The PSNR of Tubulin, CD45, and CD34 signals were, respectively, 19.59 ± 1.75 dB, 17.50 ± 2.18 dB, and 17.07 ± 1.34 dB before SpiDe-Sr enhancement, and 21.22 ± 2.33 dB, 19.51 ± 2.29 dB, and 20.92 ± 1.63 dB after SpiDe-Sr enhancement. SpiDe-Sr increased PSNR by 8.16 ± 4.77%, 11.67 ± 4.70%, and 22.60 ± 0.88% for the three markers (Fig. 3e). Meanwhile, the SSIM of the three markers raised from 0.57 ± 0.06, 0.53 ± 0.08, and 0.51 ± 0.03, to 0.72 ± 0.15, 0.67 ± 0.12, and 0.70 ± 0.07, which corresponded to increase by 27.27 ± 8.42%, 25.15 ± 5.49%, and 38.22 ± 9.47%, respectively (Fig. 3f).

The precision of cell segmentation was also improved in images enhanced by SpiDe-Sr. Using raw images of Tubulin, CD45, and CD34, the accuracy of cell extraction was, respectively, 68.24 ± 8.52%, 75.62 ± 5.92%, and 69.32 ± 4.08%. After image enhancement by SpiDe-Sr, the accuracy was increased by 23.07 ± 8.87%, 20.76 ± 12.31%, and 31.63 ± 9.96%, reaching 83.79 ± 9.76%, 90.80 ± 4.43%, and 90.98 ± 1.87%, respectively (Fig. 3g, Supplementary Fig. 5a, c). The boundaries of the extracted cells were more accurately segmented. The IoU was increased from 0.56 ± 0.18, 0.62 ± 0.13, and 0.57 ± 0.14 to 0.68 ± 0.16, 0.73 ± 0.11, and 0.68 ± 0.11, respectively, for the Tubulin, CD45, and CD34 biomarkers (Fig. 3h, Supplementary Fig. 5b). Accurate cell segmentation resulted in more precise detection of marker expressions. The expressions of Tubulin, CD45, and CD34 in, respectively, 95.32%, 98.82%, 100% of the total extracted cells became more consistent with the corresponding confocal images after SpiDe-Sr enhancement (Fig. 3i). The normalized protein expression levels in the enhanced images were also closer to confocal images than raw IMC images (Fig. 3j).

In this case, SpiDe-Sr was also compared with the other three super-resolution methods including SRCNN, KernelGAN, and RCAN.

SpiDe-Sr outperformed the other three methods in terms of PSNR, SSIM, running time (Fig. 3k, Supplementary Fig. 5d–h), and visualization (Fig. 3l, Supplementary Fig. 5c).

### SpiDe-Sr enhanced IMC images of human breast cancer tissue
To further evaluate the performance of SpiDe-Sr on human tissue samples, paired 20× confocal and IMC images of human breast cancer tissue were acquired (Fig. 4a). Cell nucleus were stained with both fluorescent (DAPI) and metal ($^{191}$Ir/$^{193}$Ir) dyes and Tubulin, CD45, and CD34 were stained with fluorescent/metal dual-labeled antibodies (Fig. 4b–d). The PSNR of Tubulin, CD45, and CD34 in raw IMC images was 17.49 ± 3.63 dB, 16.07 ± 2.05 dB, and 16.86 ± 1.70 dB, respectively. Following SpiDe-Sr enhancement, there was a notable increase in PSNR, by 17.21 ± 14.10%, 21.16 ± 11.69%, and 13.19 ± 5.97%, resulting in values of 20.10 ± 2.97 dB, 19.30 ± 1.75 dB, and 19.06 ± 1.89 dB, respectively (Fig. 4e). SpiDe-Sr enhancement also enabled elevation of SSIM of the three markers from 0.59 ± 0.08, 0.55 ± 0.09, and 0.52 ± 0.09, respectively, to 0.70 ± 0.12, 0.67 ± 0.10, and 0.71 ± 0.10, corresponding to an increase by 17.51 ± 4.50%, 20.24 ± 5.25%, and 37.27 ± 11.18% (Fig. 4f).

This enhancement correlated with increased accuracy in cell segmentation. The accuracy of cell extraction for Tubulin, CD45, and CD34 images raised to 89.84 ± 8.51%, 94.37 ± 2.31%, and 78.72 ± 10.50%, respectively, which was a noticeable increase compared to raw IMC images (Fig. 4g, Supplementary Fig. 6a), which enabled 93.85%/94.20%/92.92% of the extracted cells in the Tubulin/CD45/CD34 images being more accurately segmented. The IoU was increased from 0.56 ± 0.18 to 0.69 ± 0.16 for the Tubulin image, 0.62 ± 0.13 to 0.73 ± 0.11 for CD45, and 0.57 ± 0.14 to 0.68 ± 0.11 for CD34 (Fig. 4h, Supplementary Fig. 6b). The expressions of Tubulin, CD45, and CD34 in the IMC images also became more consistent with the paired confocal images after SpiDe-Sr enhancement (Fig. 4l, j).

Furthermore, comparative validation with other super-resolution methods like SRCNN, KernelGAN, and RCAN on human breast cancer tissue images indicated SpiDe-Sr's superior performance in terms of PSNR, SSIM, and subjective visual experience. Despite its high performance, SpiDe-Sr exhibited a relatively shorter running time of 0.44 ± 0.01 s/pic, second only to RCAN (0.33 ± 0.08 s/pic) (Fig. 4k, l, Supplementary Fig. 6c–h).

### SpiDe-Sr facilitates precise spatial proteomics analysis of breast cancer microenvironment
Bacterial colonization within the mammary gland has been reported as a crucial contributor to modulating the tumor microenvironment and impacting immunotherapeutic responses[34,35]. However, characterizing bacterial presence in tumor microenvironment remains challenging due to their typically small physical sizes[36]. Therefore, SpiDe-Sr was adopted to enhance the multiplex IMC images for higher resolution so that bacterial signals could be precisely analyzed.

We recruited a cohort of 20 patients covering 4 major breast cancer subtypes (HER2, human epidermal growth factor receptor 2 breast cancer; LA, luminal A breast cancer; LB, luminal B breast cancer; TNBC, triple-negative breast cancer) and designed a 14-channel IMC

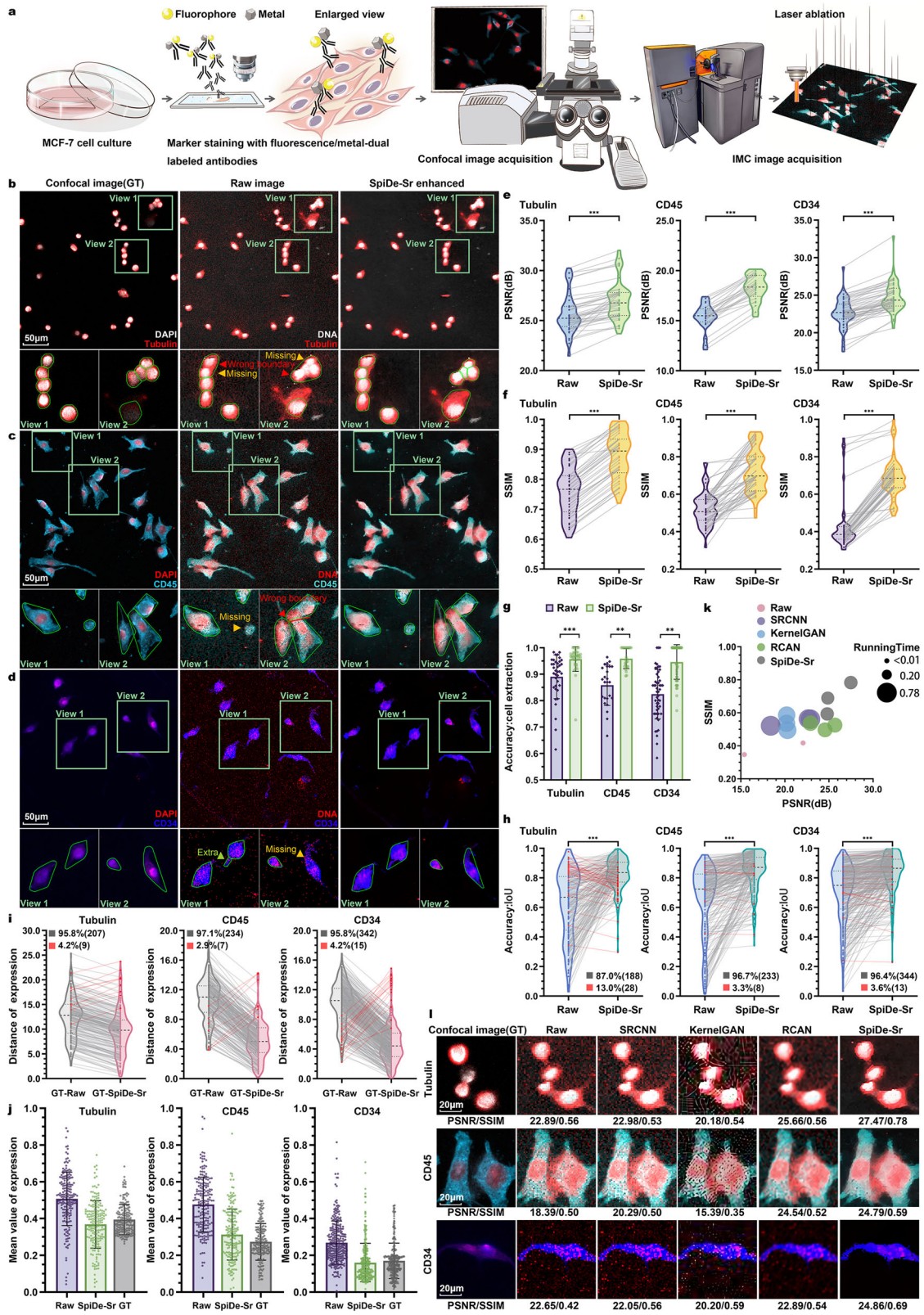

panel to simultaneously identify bacteria subtypes, breast cancer cell subtypes, and cellular functions (Fig. 5a, Supplementary Table 1). Specifically, the panel included clinically established breast cancer targets (ER, PR, HER2, ISG15, PKCD, ZC3HAV1), cell proliferation marker Ki67, apoptosis marker IFI6, immune lineage markers (CD19, CD45, CD68, CD8a), and G$^-$ (gram-negative)/G$^+$ (gram-positive) bacterial markers (LPS, LTA)[1,12,37].

After acquisition of the IMC images, SpiDe-Sr was applied to the raw images, allowing more cells to be accurately identified (Fig. 5b). Totally 269,556 cells (HER2:86,968; LA:55,496; LB:73,161; TNBC:53,931) were identified from 84 SpiDe-Sr enhanced images (Fig. 5c). The signal intensity of each cell was quantified, and normalized marker expressions of the four breast cancer subtypes were separately depicted (Fig. 5d). FlowSOM[38] was employed to determine the clusters in all

**Fig. 2 | Validation of SpiDe-Sr on IMC images of MCF-7 cell line. a** Schematic of acquiring paired images of cells with fluorescent/metal dual-labeled antibodies. **b–d** Confocal microscopy (left), raw IMC (middle), and SpiDe-Sr enhanced IMC (right) images of nucleus and examples of relatively high/moderate/low expression markers (**b** Tubulin/**c** CD45/**d** CD34). Cell segmentation was conducted with Cellpose. Missed (false negatives), extra segmentations (false positives), and wrong boundary were, respectively, indicated by yellow, green, and red arrows. Correctly extracted but wrongly bounded regions were indicated by red arrows. **e, f** Violin-scatter plots showing the distribution of (**e**) peak signal-to-noise ratio (PSNR) and (**f**) structural similarity (SSIM) with ground truth (GT) images before and after SpiDe-Sr enhancement. Each gray line represented the variation of a single image before and after enhancement. $n = 52$ (Tubulin)/36 (CD45)/71 (CD34) images. **g** Accuracy of cell extraction before and after SpiDe-Sr enhancement. Data were mean ± SD for $n = 38$ (Tubulin)/26 (CD45)/52 (CD34) images. **h** Violin-scatter plots showed the distribution of intersection over union (IoU) of accurately extracted

cells in IMC images before and after SpiDe-Sr enhancement vs. GT images. Each line represented the variation of a single cell before and after enhancement. Increasing and decreasing pairs were colored in gray and red, respectively. **i** Violin-scatter plots showed the distance of biomarker expressions in accurately extracted cells from IMC images with and without SpiDe-Sr enhancement to the corresponding cells in GT images. Each line represented the variation of a single cell before and after enhancement. Increasing and decreasing pairs were colored red and gray, respectively. **j** Normalized marker expressions in accurately extracted cells. Data were presented as mean values ± SD. In (**h–j**), the number of cells was 216/241/357. **k** Comparison of SpiDe-Sr method with the three competitive SR methods in PSNR, SSIM, and running time. **l** Visual comparison of SpiDe-Sr method with the three competitive super-resolution methods. In (**e–h**), asterisks indicate statistical significance by paired-samples two-sided t-test, **$P < 0.01$, ***$P < 0.001$. Source data are provided as a Source data file.

required cells. Compared with raw images, the clustering results were significantly improved after SpiDe-Sr enhancement. Calinski Harabasz (CH) score[39], which evaluated the degree of dispersion between clusters, was increased by 38.29 ± 24.23%, indicating the identified clusters were more discrete. Meanwhile, the Davies-Bouldin (DB) score[40], which evaluated the intra-cluster tightness, was reduced by 11.12 ± 8.73%, indicating more similarity within the identified clusters (Fig. 5e). In addition, another clustering algorithm, PhenoGraph[1,41], was also performed without any preset number of clusters. SpiDe-Sr enhancement realized 13.40 ± 3.69% increase of CH score and 6.33 ± 0.96% decline of DB score (Fig. 5f).

The clustering result of PhenoGraph with the highest CH score was used in subsequent analysis. Normal healthy cells (C1–C9), B cells (C32 and C33 with highest expression of CD19), T cells (C26 with highest expression of CD45), macrophage (C11 with highest expression of CD68), and cells containing G⁻/G⁺ bacteria (C12/C10), as well as 8 diverse tumor cell clusters were identified clearly (Fig. 5g, h). LPS and LTA were, respectively, markers of G⁻ and G⁺ bacteria. Cluster #12 (C12) with the highest expression of LPS and Cluster #10 (C10) with highest expression of LTA were further examined (Fig. 5l, j). The total numbers of cells in C12 and C10 were 847 and 3854, respectively. In C12, the LPS expression in all the four breast cancer subtypes positively correlated with tumor markers, especially IFI6, and the expression of immune markers such as CD68 was negatively correlated with LPS in all the four breast cancer subtypes except LA (lower half of Fig. 5k, Supplementary Fig. 7d and Supplementary Table 8). Inversely, in C10, the expression of immune markers, such as CD45, was positively correlated with LTA expression in all four breast cancer subtypes except LB, and the expression of LTA was negatively correlated with the expression of breast cancer markers associated with abnormal cell growth, namely HER2 and Ki67 (upper half of Fig. 5k, Supplementary Fig. 7e and Supplementary Table 9). In addition to this, we obtained label-free proteomics data of bacteria-enriched and bacteria-nonenriched regions in samples of four breast cancer subtypes. Analysis of the data revealed that proteins with significantly higher expression in G⁺ bacterial-enriched regions were associated with immunity (Supplementary Figs. 9 and 10). After analyzing the correlations, the differential expression of the proteins in C10 and C12 were also analyzed. The expression of these markers, which were highly positively or negatively correlated with LPS/LTA, were all significantly different from the expression of LPS/LTA (Fig. 5l).

Without SpiDe-Sr enhancement, B cells and T cells could not be distinguished and only 4 tumor cell clusters were identified based on the same IMC dataset because of noise interference or insufficiently precise details (Supplementary Fig. 8e, f). And in subsequent analyses, there was no indication that G⁻ or G⁺ bacteria had any particular correlation in the breast cancer microenvironment (Supplementary Fig. 8i). After SpiDe-Sr enhancement, more biological information was mined.

## SpiDe-Sr is compatible with enhancement of fluorescence microscopy images

To further exhibit the versatility of SpiDe-Sr, we tested the migration of SpiDe-Sr to conventional fluorescent images. Confocal microscope images of MCF-7 cells, mouse retina, and human FFPE breast tissues were separately acquired at 10× and 40× magnifications (Fig. 6a). The 40× images served as ground truths, and the 10× images were used as the input of super-resolution. Our findings underscored the efficacy of SpiDe-Sr in enhancing details within conventional fluorescent images across various sample types. The blur kernels estimated between the enhanced images and 10× images exhibited a high degree of similarity to the true blur kernels between the raw 40× images and 10× images (Fig. 6b–d).

For comparison, we have also tested the other three super-resolution methods (SRCNN/KernelGAN/RCAN) for the same task. In terms of subjective visual experience, SpiDe-Sr outperformed the other three methods. KernelGAN demonstrated the ability to enhance image details, however, tended to over-enhance invalid details (Fig. 6e).

Quantitative evaluation of the super-resolution results by the four methods was also performed. SpiDe-Sr demonstrated the most exceptional overall performance across all sample types, resulting in an improvement of PSNR and SSIM, respectively, by 21.08 ± 2.29% and 26.99 ± 14.04%, compared to the raw images (Fig. 6f, g). In comparison, KernelGAN led to a decline in SSIM, particularly evident in more intricate images, 0.94 ± 2.59%, 32.60 ± 17.48%, and 38.55 ± 17.83%, respectively, for MCF-7 cells, mouse retina, and human breast tissues. In terms of computational efficiency, RCAN exhibited the shortest running time of 0.37 ± 0.05 s/pic, only marginally faster than SpiDe-Sr (0.41 ± 0.05 s/pic) (Fig. 6h). In addition, F-actin were reasonably inferred by four methods in the super-resolution images, while SpiDe-Sr exhibited clearer details compared to the other three methods (Fig. 6i).

## Discussion

SpiDe-Sr integrates a blind super-resolution network with a self-supervised denoising module. The denoising module overcomes the reliance on ground truth by training a self-supervised network with image pairs that are neighbor sub-sampled from raw images. The blind super-resolution network iteratively corrects the estimated blur kernel to approach the true blur kernel in IMC, endowing the network with the capability of enhancing image without prior knowledge. In cell line, mouse tissue and human tissue samples, SpiDe-Sr rationally suppressed image noises and enhanced details, enabling more accurate cell segmentation and measurement of marker expressions at single-cell level. The specialized denoising module avoids treating noise as valid information like in other super-resolution methods. Moreover, the super-resolution network is split into three branches and trained separately, which effectively reduced the number of layers in the deep learning network. The delicate design of network structure underlays the superior performance on spatial proteomics images, and widening

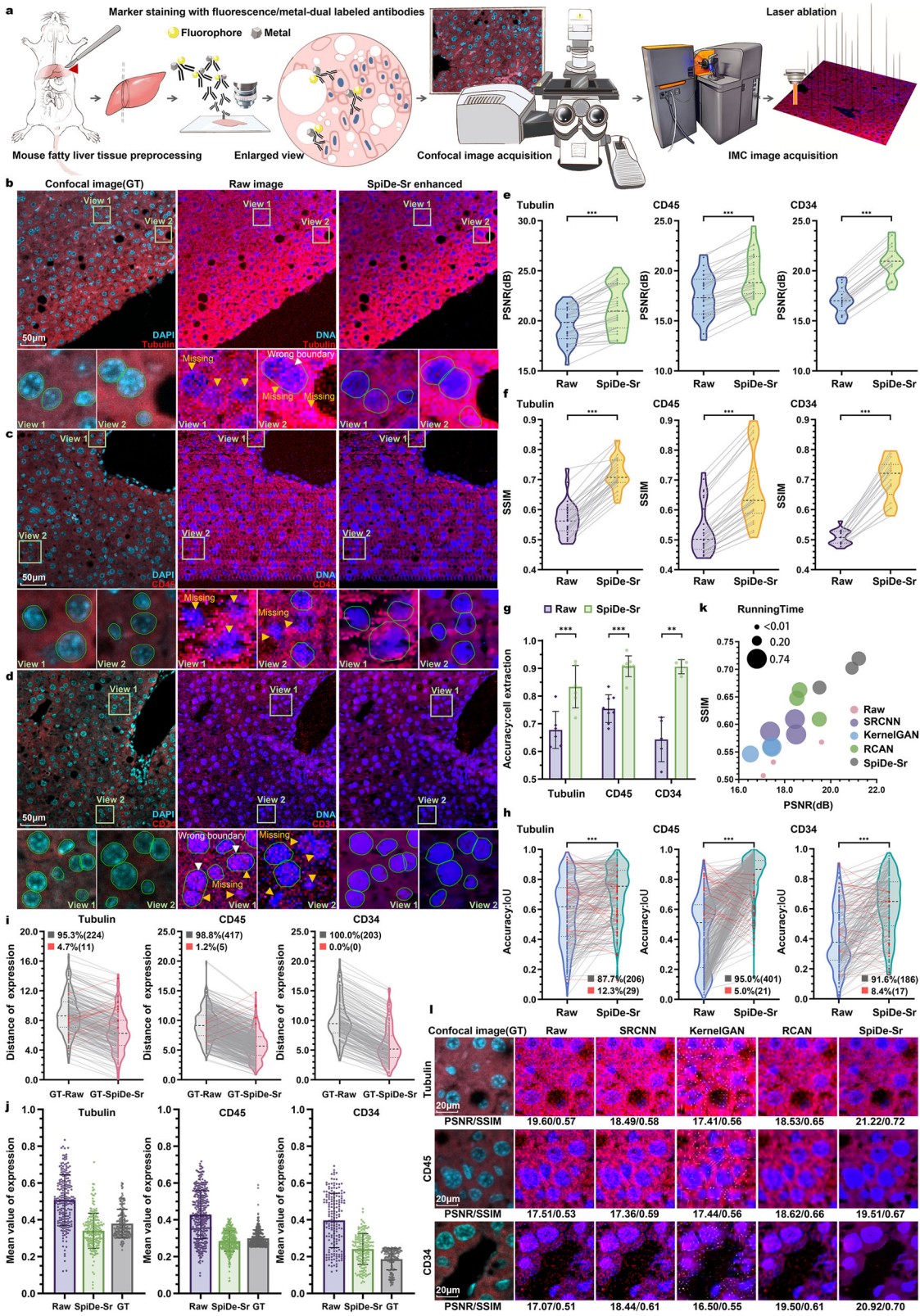

its applicability range, such as conventional fluorescence microscopy images.

In addition, SpiDe-Sr was employed to analyze IMC images from a clinical breast cancer cohort. We focused on G- and G+ bacteria relevant tumor cell cluster. Our dataset faithfully detected G- bacteria in the microenvironment, which was consistent with previous findings[34,35]. We found that the expression of G- bacteria was positively correlated

with apoptosis-related IFI6 and negatively correlated with immune-related CD68 and CD45. We speculated that the G- bacteria may have a synergistic cross-talk with IFI6. In addition, G+ bacteria have been detected in tumor tissues and exhibited a positive correlation with immune-related CD45. Thus, the presence of G+ bacteria may facilitate the inhibition of tumor proliferation. The additional analysis of label-free proteomics also cross-verified the observation that the presence

**Fig. 3 | Validation of SpiDe-Sr on IMC images of mouse fatty liver tissues.**
**a** Schematic of acquiring paired images of mouse fatty liver tissues with fluor-
escent/metal dual-labeled antibodies. **b**–**d** Confocal microscopy (left), raw IMC
(middle), and SpiDe-Sr enhanced IMC (right) images of nucleus and examples of
relatively high/moderate/low expression markers (**b** Tubulin/**c** CD45/**d** CD34). Cell
segmentation was conducted with Cellpose. Missed cells and wrong boundary
were, respectively, indicated by yellow and white arrows. Correctly extracted but
wrongly bounded regions were indicated by red arrows. **e**, **f** Violin-scatter plots
showing the distribution of (**e**) peak signal-to-noise ratio (PSNR) and (**f**) structural
similarity (SSIM) with ground truth (GT) images before and after SpiDe-Sr
enhancement. Each gray line represented the variation of a single image before and
after enhancement. $n = 37$ (Tubulin)/37 (CD45)/21 (CD34) images. **g** Accuracy of cell
extraction before and after SpiDe-Sr enhancement. Data were presented as mean
values ± SD. $n = 6$ (Tubulin)/8 (CD45)/5 (CD34) images. **h** Violin-scatter plots
showed the distribution of intersection over union (IoU) of accurately extracted

cells in IMC images before and after SpiDe-Sr enhancement vs. GT images. Each line
represented the variation of a single cell before and after enhancement. Increasing
and decreasing pairs were colored in gray and red, respectively. In (**h**–**j**), $n = 235$
(Tubulin)/422 (CD45)/203 (CD34) cells. **i** Violin-scatter plots showed the distance of
marker expressions in accurately extracted cells from IMC images with and without
SpiDe-Sr enhancement to the corresponding cells in GT images. Each line repre-
sented the variation of a single cell before and after enhancement. Increasing and
decreasing pairs were colored red and gray, respectively. **j** Normalized marker
expressions in accurately extracted cells. Data were presented as mean values ± SD.
**k** Comparison of SpiDe-Sr method with the three competitive SR methods in PSNR,
SSIM, and running time. **l** Visual comparison of SpiDe-Sr method with the three
competitive super-resolution methods. In (**e**–**h**), asterisks indicate statistical sig-
nificance by paired-samples two-sided t-test, ***$P < 0.001$. Source data are provided
as a Source data file.

---

of bacteria may modulate immune responses (Supplementary
Figs. 9 and 10).

Further improvements in SpiDe-Sr methods involve incorporating
instrumental features of IMC into model training to further advance its
blind super-resolution capabilities. Integration into user-friendly
packages for clinical researchers without algorithmic expertise is
also a goal. In addition, comparative experiments are needed to
explore the interaction mechanism of bacterial presence with immune
(CD45 and CD68) or tumor cells (IFI6 and ISG15) based on the existing
findings of marker expression correlations.

In summary, we have proposed and demonstrated SpiDe-Sr, a
method capable of denoising and enhancing resolution for mass
cytometry-based spatial proteomics imaging. Its potential applicability
to diverse clinical samples underscores its promising role in spatial
proteome research, particularly in studying tumor microenvironments
and disease pathogenesis.

## Methods
### Ethical statement
All of our experiments on mouse were ethically proved by Institutional
Animal Care and Use Committee (IACUC) of Shanghai Jiao Tong Uni-
versity (approval # 202201309) and the experiments on human samples
were ethically cleared by Institutional Review Board for Human
Research Protections of Shanghai Jiao Tong University (approval #
B2022357P). Human tissue samples were collected with previous patient
consent in strict observance of the legal and institutional regulations.

### Network architecture of the SpiDe-Sr denoising module
The denoising module was based on the self-supervised framework[25]
and trained by single observation of noisy images. The denoising
module consisted of two components, the denoising network and the
neighbor sub-sampler (Fig. 1a). The U-net[26] has been chosen by us for
spatial proteomics image denoising because it has been proposed for
biomedical images and reported to have superior performance on cell
segmentation in many studies[21,26]. Moreover, U-net as a CNN network
had a non-redundant and effective structure and possessed a relatively
comprehensive mathematical derivation compared to other newly
proposed methods. The specific U-net structure used for SpiDe-Sr was
illustrated in Supplementary Fig. 1a. Neighbor sub-sampling was pro-
posed to solve the challenging problem of capturing multiple noisy
observations of a scene in images, reducing the reliance on clean
images (ground truth) when training the denoising model. Therefore,
U-net combined with neighbor sampling was an optimal solution for
accuracy-sensitive IMC images without ground truth.

The neighbor sub-sampler ($G = (g_1, g_2)$) generated noisy image
pairs ($g_1(y), g_2(y)$) from single noise image ($y$). Noisy image pairs that
satisfy both of the following can be used for self-supervised training:
(1) The sub-sampled noisy image pairs are conditionally independent
given GT; (2) The discrepancy between the GT images of $g_1(y)$ and $g_2(y)$

is minimal[24,25]. For the raw noisy image ($y$) of spatial size M × N, the
description of $G = (g_1, g_2)$ was as follows:

Step 1: The raw noisy image ($y$) was divided into M/k × N/k cells of
size k × k. According to the experience in literature[24], set k to 2.

Step 2: In the $i$-th row and $j$-th column cell, two neighbor pixels
were randomly selected as elements of the $i$-th row and $j$-th column
pixel point in $g_1(y)$ and $g_2(y)$, respectively. $i \in [1, M/k]$, $j \in [1, N/k]$, $i, j$
were integers.

Step 3: For all cells, repeated Step 2.

Since the pixels of the paired images were neighbors in the raw
noisy image, the GT of the paired images were similar and could be
conditionally considered independent, thus satisfying the above two
conditions. Paired images with the similar ground truths were
demonstrated in the theorem proof in Supplementary Information.

### Dataset acquisition, pre-processing, and denoising module training
The IMC samples archived in our laboratory (including 91 breast cancer
samples, 67 liver cancer samples, and 63 mouse organs samples) were
prepared into 21,960 raw images of 300 × 300 pixels in TIFF format
using MATLAB scripts. These 21,960 raw images constituted a dataset
named SpiSet. Three-fifths of images in SpiSet were allocated for
training the denoising network, and one-fifth were employed for vali-
dation. The remaining one-fifth of SpiSet were randomly superimposed
with Gaussian or Poisson or pepper noise through the utilization of
built-in function within MATLAB for testing.

The loss function ($L$) employed for training the denoising network
was as follows:

$$L = L_{rec} + L_{reg} =$$
$$= \|U_\theta(g_1(y)) - g_2(y)\|_2^2 + \gamma \cdot \|U_\theta(g_1(y)) - g_2(y) - (g_1(U_\theta(y)) - g_2(U_\theta(y)))\|_2^2 \tag{1}$$

here $U_\theta$ was the U-net denoising network parameterized by $\theta$. $\gamma$ was the
hyper-parameter controlling the regularization strength, and $L_{reg}$ was
used to correct for the essential differences of ground truth pixel
values between sub-sampled noisy image pairs. The specific training
pipeline was shown in the Step 1–8 of the Pseudo code in
Supplementary Information.

In this work, the denosing network of SpiDe-Sr was trained on a
computer workstation equipped with an AMD Ryzen 5975WX CPU
running at 4.50 GHz and one NVIDIA RTX 3090 graphics processing
card, with Python version 3.7 and PyTorch version 1.7.0. We utilized a
batch size of 4 for training and Adam optimizer with the initial learning
rate of 0.0001. The number of training epochs was 100 and the
learning rate decayed by half every 20 epochs. For the hyper-
parameter $\gamma$ used to control the strength of the regularization term
was set to 1. For the configuration of the operating environment (OE),
please refer to the Supplementary Table 3.

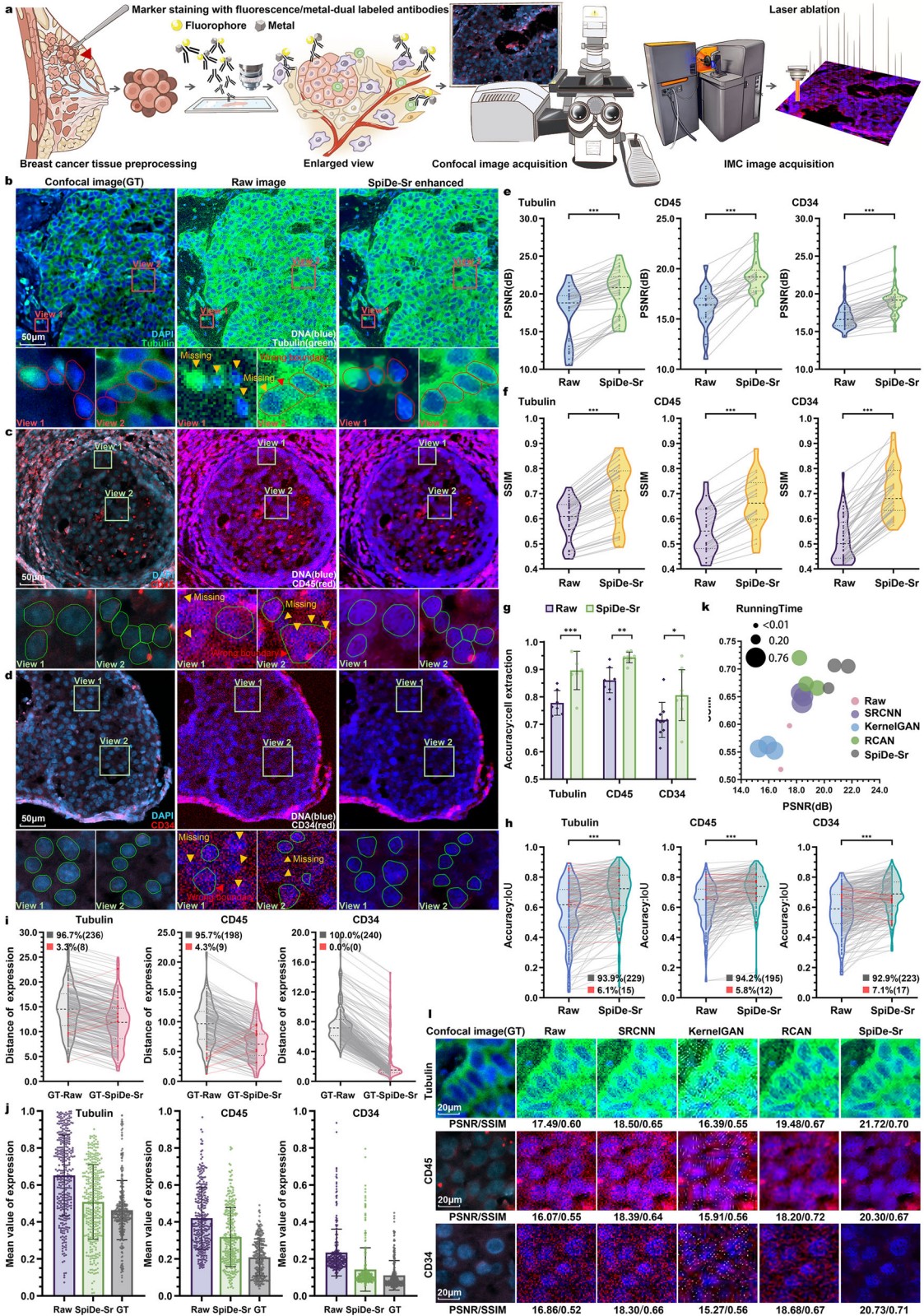

## Network architecture of the SpiDe-Sr SR module

Mathematically, the degradation model of the image is as follows:

$$I^{LR} = \left( K \otimes I^{HR} \right) \downarrow_s + n \qquad (2)$$

here $I^{HR}$ is the HR image, $I^{LR}$ is the LR image, $K$ is the blur kernel and $n$ is additional noise. $\otimes$ denotes the convolution operation and $\downarrow_s$ is the down-sampling operation[27,28]. The blur kernel is a quantitative characterization of the image degradation process. LR images are deconvoluted with matching blur kernels and then up-sampled to reconstruct high-quality HR images. When the blur kernel is unknowable, the process of reconstructing the HR image is called blind super-resolution. In blind SR studies, blur kernels are usually estimated based on specific degradation process[42]. However, the real degradation

**Fig. 4 | Validation of SpiDe-Sr on IMC images of human breast cancer tissues.**
**a** Schematic of acquiring paired images of breast cancer tissues with fluorescent/metal dual-labeled antibodies. **b–d** Confocal microscopy (left), raw IMC (middle), and SpiDe-Sr enhanced IMC (right) images of nucleus and examples of relatively high/moderate/low expression markers (**b** Tubulin/**c** CD45/**d** CD34). Cell segmentation was conducted with Cellpose. The missed cells and wrong boundary were, respectively, indicated by yellow and red arrows. Correctly extracted but wrongly bounded regions were indicated by red arrows. **e, f** Violin-scatter plots showing the distribution of (**e**) peak signal-to-noise ratio (PSNR) and (**f**) structural similarity (SSIM) with ground truth (GT) images before and after SpiDe-Sr enhancement. Each gray line represented the variation of a single image before and after enhancement. $n = 47$ (Tubulin)/25 (CD45)/54 (CD34) images. **g** Accuracy of cell extraction before and after SpiDe-Sr enhancement. Data were presented as mean values ± SD. $n = 7$ (Tubulin)/8 (CD45)/10 (CD34) images. **h** Violin-scatter plots showed the distribution of intersection over union (IoU) of accurately extracted cells in IMC images

before and after SpiDe-Sr enhancement vs. GT images. Each line represented the variation of a single cell before and after enhancement. Increasing and decreasing pairs were colored in gray and red, respectively. **i** Violin-scatter plots showed the distance of biomarker expressions in accurately extracted cells from IMC images with and without SpiDe-Sr enhancement to the corresponding cells in GT images. Each line represented the variation of a single cell before and after enhancement. Increasing and decreasing pairs were colored red and gray, respectively. **j** Normalized marker expressions in accurately extracted cells. Data were presented as mean values ± SD. In (**h–j**), $n = 244$ (Tubulin)/207 (CD45)/240 (CD34) cells. **k** Comparison of SpiDe-Sr method with the three competitive SR methods in PSNR, SSIM, and running time. **l** Visual comparison of SpiDe-Sr method with the three competitive super-resolution methods. In (**e–h**), asterisks indicate statistical significancy by paired-samples two-sided t-test, *$P < 0.05$, **$P < 0.01$, ***$P < 0.001$. Source data are provided as a Source data file.

process (that is, the true blur kernel) is complex, thus researchers have proposed to correct the estimated blur kernels to adapt them to real applications[27,43,44].

The rationality of the enhanced IMC image was the primary consideration when selecting the appropriate methods. And the methods based on predefined degradation models were more conducive to ensure the objectivity and authenticity of the resulting images. Therefore, the idea of iteratively correcting the predefined blur kernel was opted in our study for super-resolution of IMC images after denoising. Specifically, for LR image ($I^{LR}$) with dimension M×N×3 (M and N were the length and width of the image, 3 represented the three channels of RGB.), the primary procedures were outlined as follows:

Step 1: Initialize the counter $i = 0$. The initial blur kernel ($k_0$) was estimated by the predictor ($P_\theta$):

$$k_0 = P_\theta(I^{LR}) \tag{3}$$

Step 2: Input the $k_0$ and $I^{LR}$ into the SR network ($S_\theta$) and output the first SR image ($I^{SR}_0$). The blind SR network employed the SFTMD, which avoided the image-independent interference that would be introduced by processing the blur kernel and the LR image simultaneously with the convolution operation[28].

$$I^{SR}_0 = S_\theta(I^{LR}, k_0) \tag{4}$$

Step 3: Update counter $i = i + 1$. The blur kernel was iteratively corrected with the corrector ($C_\theta$) as follows:

$$\Delta k_i = C_\theta(I^{SR}_{i-1}, k_{i-1}) \tag{5}$$

$$k_i = k_{i-1} + \Delta k_i \tag{6}$$

Here, $\Delta k_i$ was the error between the true blur kernel ($K$) and the predicted blur kernel at the $i$-th iteration. $k_i$ was the output blur kernel after the $i$-th correction and $k_{i-1}$ was the previous output of the $i$-th.

Step 4: Input the corrected blur kernel ($k_i$) and the $I^{LR}$ into SFTMD, and output the $i$-th SR image ($I^{SR}_i$):

$$I^{SR}_i = S_\theta(I^{LR}, k_i) \tag{7}$$

Step 5: Repeat Step 3 and Step 4 until the model converges.

### Dataset acquisition, pre-processing, and SR module training
The images in SpiSet were augmented with random horizontal flips and 90 degrees rotations to obtain the HR images. The isotropic Gaussian blur kernel with width range set to 0.2 to 4.0 and size fixed to 21*21 was employed as blur kernel in our work ($K$). For non-moving images, the isotropic Gaussian blur kernel has been widely adopted in

previous studies[27,28]. The width of the blur kernel was the standard deviation of the Gaussian function ($\sigma \in [0.2, 4.0]$). The HR images were convolved with the blur kernel and then down-sampled by bicubic interpolation to generate the LR images, forming HR-LR image pairs. These image pairs and their corresponding predefined blur kernel were divided into training set, validation set, and test set in the ratio of 6:2:2. For testing or validation, bicubic interpolation was used to align the image sizes when the SpiDe-Sr was not required.

The three branches of the SR module were trained on the training set (Fig. 1a and Supplementary Fig. 2a–c). First, the SR network (SFTMD[27]) was trained with mean square error (MSE) loss and then the trained parameters were fixed. Next, the predictor ($P_\theta$) and the corrector ($C_\theta$) were trained alternately. The predictor was optimized by the following formula:

$$\theta_p = \arg_{\theta_p} \min \|K - P(I^{LR}; \theta_p)\|_2^2 \tag{8}$$

Here, $\theta_p$ was the hyper-parameter of the predictor $P_\theta$. $K$ was the predefined true blur kernel. And the corrector was optimized by the following formula:

$$\theta_c = \arg_{\theta_c} \min K - (C_\theta(I^{SR}; \theta_c) + k_{i-1})_2^2 \tag{9}$$

Here, $\theta_c$ was the hyper-parameter of the predictor.

The specific training pipeline was shown in the Step 10–17 of the Pseudo code in Supplementary Information. On the validation set, the model converged by the 9-th iteration. After 9 iterations, the $I^{SR}_9$ was the final output of SR module. For additional inference processes, please consult the relevant literature[27,28]. The optimizer employed Adm with $\beta_1 = 0.9$, $\beta_2 = 0.999$, and the learning rate was set to 0.0001. The SR module was implemented with the PyTorch framework and the hardware configuration used for training the denoising module was utilized.

### Sample preparation
**Cell line sample preparation.** MCF-7 cells (HTB-22, ATCC) were grown in Dulbecco's modified Eagle's medium, containing 10% fetal bovine serum, and 1% penicillin-streptomycin. To obtain adherent cell sample, MCF-7 cells were seeded at 96-well plates with a density of ~10,000 cells per well overnight at 37 °C with 5% $CO_2$.

**Mouse liver sample preparation.** Wild-type C57BL/6J mice around 6 weeks were used in this study (Sex was not considered in our study design). Mouse fatty liver tissues were formalin fixed and paraffin embedded (FFPE), sectioned at a thickness of 5 μm, and mounted on positively charged slides to prevent tissue detachment during processing.

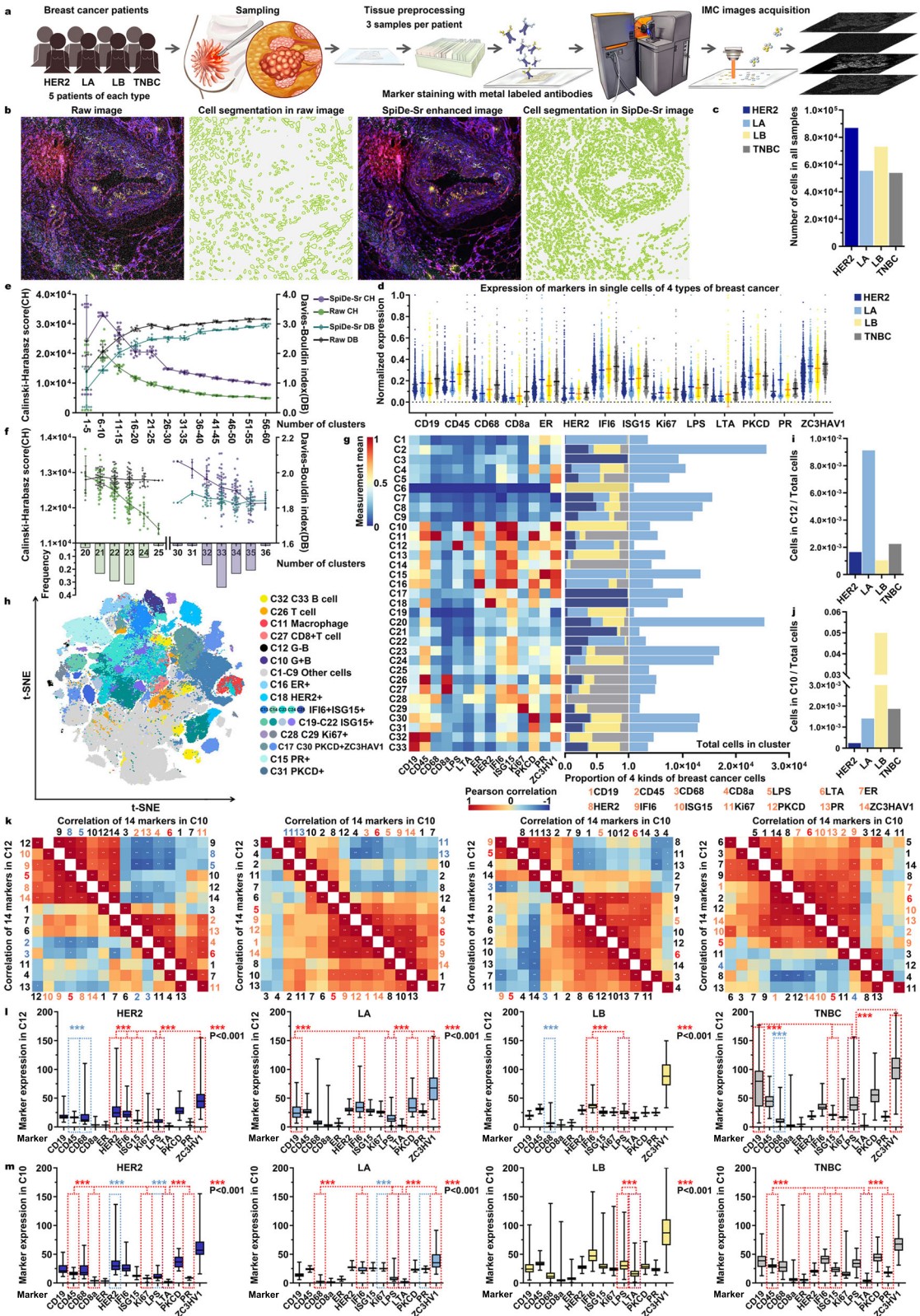

**Human breast cancer sample preparation.** Thirty-one FFPE samples of human breast cancer archived in our laboratory were from Xinhua Hospital (not duplicated in SpiSet), and were utilized for fluorescent/metal-dual-labeling experiments. The clinical breast cancer cohort (20 FFPE samples of breast cancer) was provided by Wenling First People's Hospital and identified by doctor Yuli Hu.

**Mouse retina cryo-sections preparation.** Wild-type C57BL/6J mice around 6 weeks were used in this study (Sex was not considered in our study design). Mouse retina cryo-sections were made of freshly harvested eyes. The eyes were briefly washed in PBS and fixed in 4% w/v paraformaldehyde (PFA) for 1 h. Following dissection, retinas were immersed in 4% PFA containing 30% sucrose overnight. After drying,

**Fig. 5 | Application of SpiDe-Sr to spatial proteomics data from four major subtypes of breast cancer patients. a** Workflow of IMC image acquisition. **b** A raw breast cancer IMC image and extracted cells (left), compared with the corresponding SpiDe-Sr enhanced image and extracted cells (right). **c** Number of cells. **d** Normalized expressions of 14 markers at single-cell level ($n = 8697/5550/7316/5393$ for HER2/LA/LB/TNBC). **e** CH and DB scores of FlowSOM clustering results. **f** CH and DB of PhenoGraph clustering results. Histogram showed the frequency distribution of cluster numbers. In (**d–f**), data were presented as mean ± SD. In (**e–f**), both CH and DB were statistically different before and after SpiDe-Sr enhancement (two-sided t-test, $P < 0.001$). **g** The clustering result with the highest CH score. The heat map (left) showed normalized mean marker expressions of each cluster. The stacked bar plot showed (middle) the proportions of four breast cancer cells in each cluster. The bar plot (right) showed the absolute cell counts in each cluster. **h** t-SNE (t-distributed stochastic neighbor embedding) map of 269,556 cells sub-sampled from all images. Cell clusters were marked by different colors. **i, j** The proportion of cells of each breast cancer in the clusters which had the highest expression of G⁻ (**i**, C12, $n = 144/506/76/121$ cells) and G⁺ (**j** C10, $n = 21/78/3654/101$ cells) bacterial markers, compared to the total cell count of each subtype. **k** (below) Heat map showing the Pearson correlation coefficients of the 14 markers in C12 with each other. (above) Heat map showing the Pearson correlation coefficients of the 14 markers in C10 with each other. Positively and negatively correlated markers were colored in orange and blue, respectively. LPS and LTA were colored in red. **l–m** Box plots showed the absolute expressions of 14 markers in C12 (**l** $n = 847$ cells) and C10 (**m** $n = 3854$ cells) of the four breast cancer subtypes. Red and blue asterisks, respectively, represented the statistical significance of proteins positively and negatively associated with LPS/LTA versus LPS/LTA (two-sided t-test, **$P < 0.01$, ***$P < 0.001$). Source data are provided as a Source data file.

retina was snapped frozen in OCT and sectioned at a thickness of 20 μm in crtostat (CryoStar NX50, Thermo Fisher Scientific, USA).

### Antibody preparation

Metal-labeled primary antibodies and fluorophore/metal-dual-labeled secondary antibodies were obtained using the Maxpar antibody labeling kit[45]. Of note, the secondary antibody used here was already labelled with Alexa Fluor 488 fluorophore and preserved in carrier/protein-free buffer. After conjugation, the metal-labeled antibodies were diluted in protein stabilizing cocktail for long-term storage at 4 °C. Antibodies, clones, vendors, catalog numbers, and the concentrations used in this study were listed in Supplementary Table 1.

### Immunostaining

Before immunostaining, adherent cells were briefly washed with PBS and fixed with 4% w/v PFA in PBS buffer for 10 min, followed by washing with PBS three times. FFPE samples, including mouse liver section and breast cancer section, were baked at 55 °C for 30 min, followed by deparaffinization in 100% xylene for 20 min, and rehydrated by ethanol series (100%, 95%, 80%, 70%) for 5 min each. The samples were incubated in antigen retrieval buffer and placed in an autoclave (pre-heated to 95 °C) at 95 °C for 30 min. Slides were allowed to cool to room temperature for 60 min, followed by two washes of 10 min in ddH$_2$O and PBS. As for frozen samples, mouse retina cryosections were taken out from −20 °C and equilibrated to room temperature for 1 h.

Step 1: Samples were incubated with permeabilization/blocking buffer (1× PBS containing 0.1% v/v Triton X-100 and 3% w/v BSA) for 30 min.

Step 2: Slides were incubated with primary antibodies at the appropriate concentrations (Supplementary Table 1) overnight (>8 h) at 4 °C.

Step 3: For confocal/IMC imaging, samples were incubated with fluorophore/metal-dual-labeled secondary antibodies for 1 h. The nucleus was stained with DAPI at 1:1000 dilution (1 μg/mL) for 10 min for nuclear confocal image acquisition. Correspondingly, the nucleus was stained with $^{191}$Ir/$^{193}$Ir DNA intercalator at 1:400 dilution (312.5 nM) for nuclear IMC image acquisition. For IMC imaging only, specimens were stained with $^{191}$Ir/$^{193}$Ir DNA intercalator at 1:400 dilution (312.5 nM) after primary antibodies incubation. For specific information on all reagents used in our work, please refer to the Supplementary Table 2.

### Image acquisition

Confocal images were acquired on a confocal microscope (LSM 800, Zeiss, German) and saved as 16-bit TIFF images in the ZEN blue 3.3 (Zeiss, German). The images shown in Figs. 2–4 and Supplementary Figs. 3–6 were acquired with 20×/0.40 NA LD PlnN objective. The samples used in Fig. 6 were imaged using 10×/0.3 NA EC PlnN objective and 40×/0.6 NA LD PlnN objective. All IMC images were acquired using a Hyperion laser scanning module coupled to Helios mass cytometer

(Fluidigm Sciences)[46]. A metal-coated tuning slide (Fluidigm Sciences) was used for optimization of peak intensity and resolution as a function of helium and argon flow. To minimize batch-to-batch variance, a standard internal metal isotope bead was acquired with samples together as a normalization guideline. The acquired raw data was displayed and initially analyzed in MCD Viewer (Fluidigm Sciences) and then saved as 16-bit TIFF images. Then Confocal images were paired with IMC images of the same sample using MATLAB (MATLAB 2019b) scripts.

### Data analysis of clinical breast cancer cohort

The data processing pipeline was consistent with the standard processing pipeline steps at https://github.com/BodenmillerGroup/ImcSegmentationPipeline, except that the methods in the individual steps were changed to those that performed better in the researches. The specific processing was as follows:

Step 1: The raw data were imported and displayed in the software (MCD Viewer, Fluidigm), and the valid marker channel of the raw data was selected by an experienced researcher and then stored as 16-bit TIFF format.

Step 2: A customized MATLAB script was utilized to collate all images so that the content on each image was an overlay of the nucleus channel and one marker channel. There were 14 markers in each ROI, and 14 images were saved out. The nucleus served primarily for localization.

Step 3: All images after collation were super-resolved with SpiDe-Sr.

Step 4: The regions of individual cells in all images were segmented at the pixel level using cytoplasm pattern with adaptive calibration diameter in Cellpose to generate masks. Other default parameters were in Supplementary Table 10. The mask for single-cell segmentation in each ROI was manually adjusted and selected. Single-cell segmentation mask and TIFF images of the 14 channels were overlaid to extract the average expression of markers and spatial features (cell area, perimeter, long-axis length, and short-axis length) of single cell using the MATLAB toolbox regionprops. Single-cell marker expressions were summarized by mean pixel values for each channel. The single-cell data were censored at the 99-th percentile to remove outliers, and normalized to the 99-th percentile, as was suggested for these algorithms[41,47].

Step 5: Single cells from clinical cohorts were clustered into groups with functionally similar using two unsupervised clustering methods, FlowSOM and PhenoGraph. Both methods were implemented using the python package in the download path provided in the literature[38,41]. The FlowSOM was repeated 10 times using default parameters within each determined cluster number interval. Every 5 clusters were set as one interval, for a total of 12 intervals between 1 and 60 of cluster numbers. The PhenoGraph was used for the case where the clustering number was not determined, and was repeated 120 times with the nearest neighbor parameter of 30.

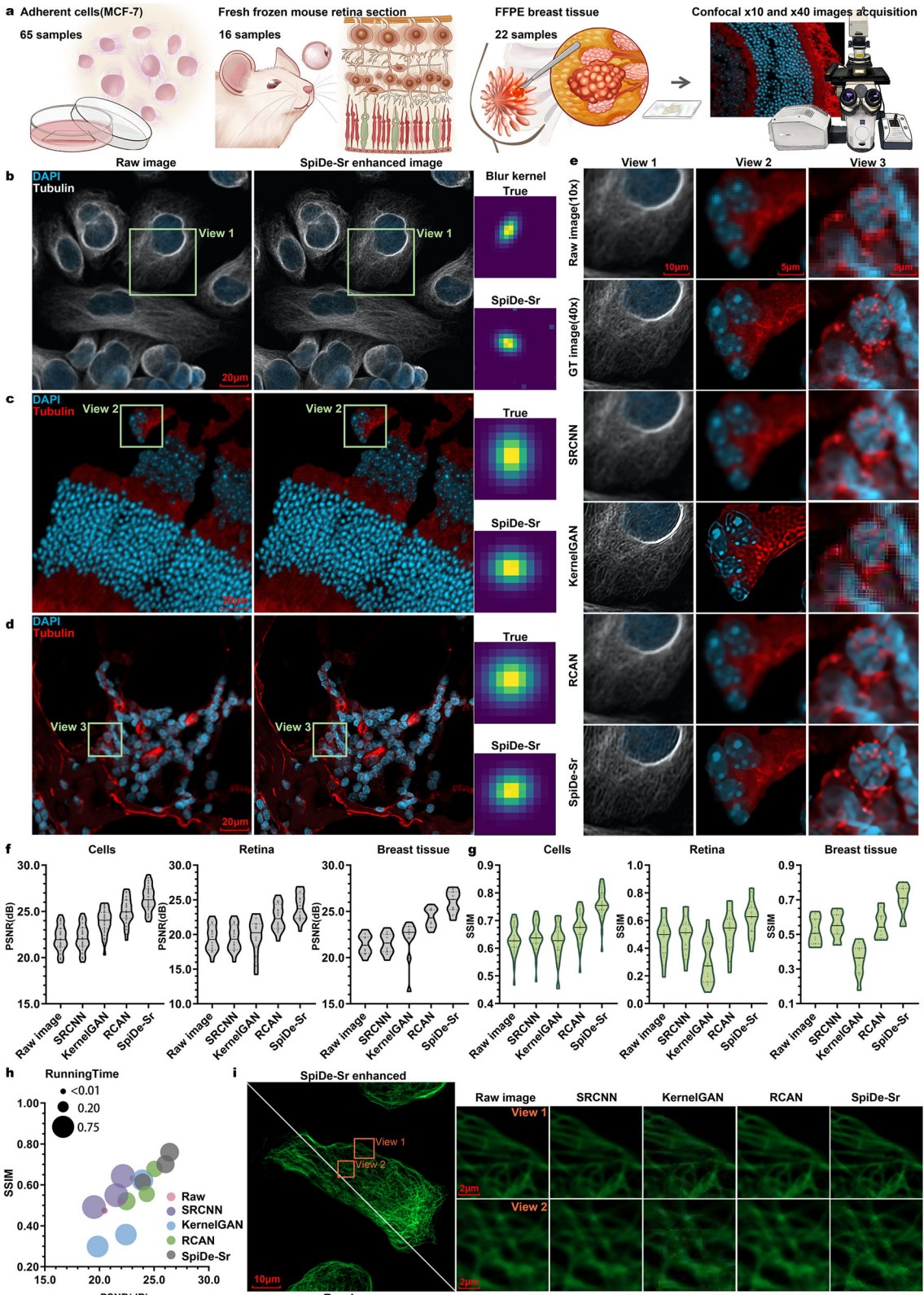

**Fig. 6 | Migrating SpiDe-Sr to fluorescence microscopy images. a** Paired images at different magnifications (10× and 40×) were acquired for MCF-7 cell line, mouse retina, and breast tissue. **b, d** Raw images at 10× magnification (left) of MCF-7 cell (**b**), mouse retina (**c**), and breast tissue (**d**), and corresponding 40× images reconstructed from the 10× images using SpiDe-Sr (middle), along with the true blur kernels and the blur kernels between the 10× and SpiDe-Sr enhanced 40× images (right). **e** Comparison of the super-resolution images reconstructed by SRCNN, KernelGAN, RCAN, and SpiDe-Sr for the three sample types. **f, g** Comparisons of

PSNR (**f**) and SSIM (**g**) among the four super-resolution methods in different sample types. n = 65/16/22 for MCF-7 cells/mouse retina tissues/human breast tissues. **h** Overall comparison of the PSNR, SSIM, and running time among the four super-resolution methods. **i** Visual comparison of 40× ground truth image of F-actin and 40× super-resolution image reconstructed from 10× image using SpiDe-Sr, as well as the other three methods. FFPE formalin fixed paraffin embedded. Source data are provided as a Source data file.

Step 6: The clustering results with the highest CH in Step 5 were used for the subsequent analysis. Functionally similar clusters were aggregated into larger groups based on the expression and correlation of markers. For visualization, high-dimensional single-cell data were reduced to two dimensions using the nonlinear dimensionality reduction algorithm t-SNE[47].

Step 7: Statistical analysis (correlation, difference) was performed on clusters with high expression of G$^-$ and G$^+$ bacteria.

For data analysis without SpiDe-Sr (in Supplementary Fig. 8), the processing pipeline was all the steps above except Step 3.

## Comparison of SpiDe-Sr and other SR methods

To validate the superiority of SpiDe-Sr, we quantitatively evaluated the performance of SpiDe-Sr and other SOTA SISR (single-image super-resolution) methods on seven different datasets: the IMC test set (in Fig. 1), fluorescent/metal-dual-labeling experimental images of cells/mouse/human (in Figs. 2–4), and conventional fluorescence microscopy images of cells/mouse/human (in Fig. 6), respectively. HR and LR images with the same scene were paired in all above datasets.

The methods used for comparison were SRCNN[30], KernelGAN[31], and RCAN[20,32], respectively. SRCNN/KernelGAN/RCAN was the top-performing CNN/GAN/Attention method in SISR task, and was used for comparison with optical microscopy methods[19,20].

Specifically, the three comparison models were retrained separately in our SR training set according to the standard procedure and optimal parameters in the research papers[30–32]. For all types of biological samples, only one model was trained to process images of different structural features and marker expressions. The super-resolved images were resized using nearest neighbor interpolation to match the dimensions of the HR image, enabling quantitative evaluation parameters to be calculated.

## Performance metrics

Two metrics were utilized to quantitatively evaluate the performance of SpiDe-Sr in enhancing image quality. PSNR and SSIM were used to evaluate pixel-level similarity between IMC images and ground-truth images, with PSNR focusing on noise levels and SSIM focusing on structural details[20,21]. In fluorescent/metal dual-labeling experiment, confocal images were utilized as the ground truth for comparison with IMC images of cells/mouse/human tissue samples before and after enhancement. PSNR and SSIM between the confocal image $x(i,j)$ and the IMC image $y(i,j)$ are calculated as:

$$\text{MSE} = \frac{1}{m \bullet n} \bullet \sum_{i=0}^{m-1} \sum_{j=0}^{n-1} [x(i,j) - y(i,j)]^2 \tag{10}$$

$$\text{PSNR} = 10 \bullet \log_{10} \left[ \frac{(2^n - 1)^2}{\text{MSE}} \right] \text{(dB)} \tag{11}$$

$$\text{SSIM} = \frac{(2 \bullet \mu_x \bullet \mu_y + \varepsilon_1) \bullet (2 \bullet \vartheta_{xy} + \varepsilon_2)}{(\mu_x^2 + \mu_y^2 + \varepsilon_1) \bullet (\vartheta_x^2 + \vartheta_y^2 + \varepsilon_2)} \tag{12}$$

Here, $m$ and $n$ are the length and height of the image, $i$ and $j$ are the corresponding pixel points; $\mu_x$ and $\mu_y$ are the mean values of image $x$ and $y$, respectively; $\vartheta_x$ and $\vartheta_y$ are the variances of image $x$ and image $y$, respectively; $\vartheta_{xy}$ is the covariance of $x$ and $y$. $\varepsilon_1$ and $\varepsilon_2$ are two default constants of 6.5025 and 58.5225, respectively. In experiments, we calculated the PSNR and SSIM for each of the three RGB channels separately taking the average value.

Next, we also evaluated the performance of SpiDe-Sr on the basis of more complex downstream tasks such as cell segmentation and intracellular protein expression detection, which were the most crucial prerequisites in functional analysis of single-cell spatial proteomics

data. Cell extraction was regarded as an instance segmentation problem, accuracy and object-level metrics (IoU and $F_1$) were adopted to evaluate the segmentation performance of Cellpose[29] before and after enhancement. Further details were in Supplementary Note 7 and Supplementary Table 10. The accuracy is calculated as:

$$\text{Accuracy} = \frac{TP}{2 \bullet TP + FP + FN} \tag{13}$$

Here $TP$, $FP$, and $FN$ are the cell number of true positives (accurately detected cells), false positives (extra cells), and false negatives (missing cells), respectively.

The precision of the extracted cell boundaries was evaluated using IoU (intersection over union) and $F_1$. IoU is defined as the intersection area divided by the union area of two objects, and is calculated as:

$$\text{IoU} = \frac{Area_{cell1} \bigcap Area_{cell2}}{Area_{cell1} \bigcup Area_{cell2}} \tag{14}$$

Here $Area_{cell1}$ is the area of the cell that is accurately detected in the IMC image, and $Area_{cell2}$ is its area in the corresponding confocal image. $F_1$ is the pixel-level statistical complement of IoU and is calculated as:

$$F_1 = \frac{2 \bullet TP}{2 \bullet TP + FP + FN} \tag{15}$$

Here $TP$, $FP$, and $FN$ are the number of true positives, false positives, and false negatives of the pixel points of the accurately detected cells, respectively. The accuracy of intracellular protein expression detection was evaluated by the distance and mean value of the pixel values in each cell in IMC image versus corresponding cell in confocal image.

Cell clustering was the task that followed cell segmentation in single-cell proteomics data analysis. Furthermore, we evaluated the impact of SpiDe-Sr on cell clustering task on clinical cohort data. Calinski-Harabaz (CH) score and Davies-Bouldin (DB) score were used to evaluate the results of clustering[39,40]. The CH is defined as the ratio of the inter-cluster distance to the intra-cluster distance, and DB measures the similarity between each cluster and its most similar clusters. The n-dimensional dataset is clustered into $k$ clusters, CH and DB are calculated as:

$$\text{CH}(k) = \frac{tr(B_k) \bullet (n - k)}{tr(W_k) \bullet (k - 1)} \tag{16}$$

$$\text{DB}(k) = \frac{1}{k} \sum_{i=1}^{k} \max_{i \neq j, j \in [i,k]} \frac{S_i + S_j}{M_{ij}} \tag{17}$$

Here n is the number of samples, $k$ is the number of clusters, $B_k$ is the inter-cluster covariance matrix, $W_k$ is the intra-cluster covariance matrix, and tr is the trace of the matrix. In the formula for DB, $i$ and $j$ are the $i$-th and $j$-th clusters, respectively. $S_i$ is the average distance of individuals in the $i$-th cluster to the center. $S_j$ is the average distance of individuals in the $j$-th cluster to the center. $M_{ij}$ is the distance between the centers of the two clusters of the $i$-th and $j$-th clusters. Max is the maximum value.

The evaluation process was implemented with customized MATLAB R2019b scripts, PSNR, SSIM, IoU, $F_1$, mean value, CH, DB, and Pearson correlation coefficient were computed using built-in functions. And the running time of the program for processing each image was obtained from the built-in timing function of PyCharm 2020.3.3.

## Statistics and reproducibility

The violin plot is a combination of the standard Tukey box-and-whisker plot and density distribution plot, showing the distribution of datasets as well as probability densities. The three lines in the violin plot represent the upper quartile, median, and lower quartile, respectively. All violin plots (in Figs. 2–4e, f, h, i and Fig. 6f) were plotted in GraphPad Prism 9 in the standard format, and we superimposed the scatter plot of the data on top of it after aligning the coordinates. The asterisk in the violin plots indicated the statistically significant difference between the two arrays as determined by two-sided paired-samples t-test. The two-sided t-tests were done in IBM SPSS Statistics 25 following standard procedure. In addition, all histograms and bubble plots were generated in GraphPad Prism 9 in the standard format. The heat map (in Fig. 5g, k) was performed using the OmicStudio tools at https://www.omicstudio.cn/tool following the advanced heat map process. In Fig. 5k, Pearson correlation coefficients greater than 0.75 were marked with two asterisks and greater than 0.5 were marked with one asterisk. In Fig. 5l, m, the center line, box limits, and whiskers of Box plots indicate the median, upper and lower quartiles and 1.5× interquartile rage. Each experiment in Fig. 2b–d, Fig. 3b–d, Fig. 4b–d, Fig. 6b–d was repeated independently 10 times with similar results. In Fig. 5e, the clustering was repeated 10 times for each cluster number interval. In Fig. 5f, the clustering was repeated 120 times without preset cluster number.

## Reporting summary

Further information on research design is available in the Nature Portfolio Reporting Summary linked to this article.

## Data availability

Source data are provided as a Source data file. The proteomics raw data of clinical breast cancer cohort used in this study are available in the the ProteomeXchange database under accession code PXD050123. Source data are provided with this paper.

## Code availability

The laboratory version of the code was published on https://github.com/DingLabSJTUChenRui/SpiDe-Sr. (https://doi.org/10.5281/zenodo.10669093).

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

## Acknowledgements

This work was supported by National Key R&D Program of China (2022YFC2601700, 2022YFF0710202) and NSFC Projects (T2122002, 22077079, 81871448), Shanghai Municipal Science and Technology Project (22Z510202478), Shanghai Municipal Education Commission Project (21SG10), Shanghai Jiao Tong University Projects (YG2021ZD19, Agri-X20200101, 2020 SJTU-HUJI), Shanghai Municipal Health Commission Project (2019CXJQ03). Thanks for AEMD SJTU, Shanghai Jiao Tong University Laboratory Animal Center for the supporting.

## Author contributions

X.D. and R.C. conceived the idea. X.D. supervised the research. R.C. and J.X. designed the experiments. J.X. prepared samples and performed experiments. R.C. and B.W. analyzed the data. R.C. wrote the manuscript, with input from all authors. Y.D. and A.A. drew the figures in Supplementary Information. Y.L. guided the use of the IMC instrument. L.J. provided the clinical samples. All authors discussed the results and commented on the manuscript.

## Competing interests
The authors declare no competing interests.
