## [Peer Review File · Nature Communications]

Reviewers' Comments:

Reviewer #1:

Remarks to the Author:

The article presents a method for post-processing images from a technique called Imaging Mass Cytometry (IMC). This post-processing primarily involves the removal of unknown noise and up-sampling of the images. A major challenge is that the distribution of noise and the down-sampling degradation model are unknown. For noise, this paper employs a method similar to Noise2noise, and uses a sub-pixel shuffle method to create training data from noisy data. This is a clever approach. It works well when the noise is independently sampled for each pixel, but may encounter issues if the noise has spatial correlations. Fortunately, the examples shown in the article demonstrate that this method is suitable for IMC images. Regarding up-sampling, an older algorithm called IKC is utilized. This method was introduced in 2019 (which is considered old in the rapidly advancing field of computer vision. IKC is likely the first deep learning-based blind SR method). Although it seems outdated, I think the selection of IKC is quite appropriate. Firstly, I believe the SR of IMC images is different from the general SR research in computer vision. For natural images, visual quality is the priority. Therefore, many Blind SR methods have emerged post-IKC to enhance visual quality, such as Real-ESRGAN and BSRGAN. However, these come at the cost of objectivity in the resulting images. For the processing of IMC images proposed in this paper, these methods are not suitable. Instead, methods based on pre-defined degradation models are more conducive to ensuring the objectivity and authenticity of the resulting images. I didn't see relevant discussions in the article, but I believe these discussions are important.

From a computational standpoint, the method presented in this paper has lower innovation, but is reasonable and effective. It's hard for me to evaluate the application value as I am not familiar with IMC and its related industries and applications. I can see that the method improves the accuracy of subsequent detection and segmentation, but I am not quite sure about the real-world value of these numerical improvements.

Reviewer #2:

Remarks to the Author:

Chen et al. proposes a new algorithm, called SpiDe-Sr, for denoising of IMC images with the goal of improving the assay resolution. The SpiDe-Sr potentially addresses a very important limitation of IMC data analysis. However, the evaluation of manuscript in its current form is almost impossible due several reasons, some of which are noted here. I hope these comments would help the authors in revising their manuscript.

The major issue is that Che et al. fails to clearly communicate its findings, especially for broad readership of Nature Communications.

There are numerous grammatical mistakes throughout the manuscript. Several figure panels are not discussed (why are they include in the first place?)

In summary, it is utterly impossible to evaluate the scientific merit of the manuscript given its poor presentation. Some of the sentences are incomprehensible. There are many unnecessary abbreviations (do the authors really need to abbreviate "state-of-the-art" to SOTA?). On the other hand, there are abbreviations with unclear definition (e.g. LA, LB, TNBC, etc.)

Performance metrics are not defined. What is SSIM? What is pSNR? How "accuracy of cell extraction" is defined / calculated Definition of terminologies/ metrics / etc should be clarified, especially for a journal such as Nature Communications with a broader readership. A short introduction of methods included in comparative analysis, tools used in throughout the paper, metrics used for evaluation, etc. would help potential readers (as well as reviewers). This is not an IEEE journal.

Here are a non-exhaustive list of examples that hopefully could guide authors in their revision.

What do authors mean by "the same underlying scene were required", given imaging the same section used for IMC is impossible? How "same" is determined?

Figures are unclear. The legends are missing. Considering Figure 3 as an example, what is GT? What do red / gray legends in panel i represent? In the same panel, what are those numbers? This is just an example. Similar problems exist throughout the manuscript.

What is the difference between "penetrating" and "existing" tumor microenvironment studies? By the way, what is the meaning of penetrating analysis?

Line 588: what do authors mean by "8 diverse tumor cell phenotype"? How do they know the phenotype of cells?

IFI6 is just a marker of apoptosis. What do the author mean by stating that "correlated with tumor makers, especially IFI6" (Line 596)?

What do the authors mean by "associated with malignant appreciation, namely HER2 and Ki67" (Lines 600-601). How malignancy can be appreciated?

The authors stated that "The most straightforward strategy to obtain IMC images with high PSNR is to manually eliminated pixel values above and below empirical thresholds". How are empirical thresholds chosen? A brief explanation is needed.

At line #216, the authors stated that "In addition, SpiDe-Sr was visually superior to three state-of-the-art (SOTA) single image SR methods including SRCNN, KernelGAN and RCAN". Quantitative superiority should be established by showing results from "SOTA" SR methods in Figures 1c, d, e and all other figures, where breast tumors and mouse samples are discussed.

Where did the authors get the Ground truth of cell segmentation? In other words, how exactly is it known that which cell "extraction" (which the reviewer is unsure of its meaning) is correct/missing/extra when the sample is analyzed by Cellpose?

How does SpiDe-Sr compare with original images which were down-sampled (lines 207-208).

Many details of methods and procedures are missing. For instance, what parameters were used for Cellpose. Did the authors attempt to train a model for Cellpose (which is obviously possible and should be done). This would be a more accurate comparison.

Why Field of View 1 is chosen in Figure 1 1f and View 2 chosen for fig 1g even though View 1 is larger than View 2?

The authors stated that "All methods except SpiDe-Sr treated the CD8 (red pixel points) that was artificially judged as not being expressed in View 2 as effective information". What is meant by artificially judged here?

The authors should visualize the learnt mappings by the networks for interpretability and explanation of the results.

Line #207, raw images were downsampled to 1/4th of original size. Why was this ratio chosen? Were other ratios tried by the authors? Preliminary results on the same can be included.

There is no discussion on the statistical significance of the improved results.

Besides technical ambiguities, I have concerns about the biological data presented in the manuscript. For example, would authors explain why leukocyte marker CD45 can be detected in MCF7 breast cancer cells?

Immune cells and epithelial cells are morphologically very different. Could the authors, use two cell

types with more similar morphology as the basis of their experiments. Lets say CD8 vs CD19?

RESPONSES TO REVIEWER COMMENTS (NCOMMS-23-42220-T)

Reviewers' comments are in *blue*, with authors' responses immediately following each Reviewer's comment and in *black*. Modifications to the manuscript are highlighted with a **yellow** background. Page numbers refer to the revised manuscript.

Response to Comments from Reviewer #1

Reviewer #1:

The article presents a method for post-processing images from a technique called Imaging Mass Cytometry (IMC). This post-processing primarily involves the removal of unknown noise and up-sampling of the images. A major challenge is that the distribution of noise and the down-sampling degradation model are unknown. For noise, this paper employs a method similar to Noise2noise, and uses a sub-pixel shuffle method to create training data from noisy data. This is a clever approach. It works well when the noise is independently sampled for each pixel, but may encounter issues if the noise has spatial correlations. Fortunately, the examples shown in the article demonstrate that this method is suitable for IMC images. Regarding up-sampling, an older algorithm called IKC is utilized. This method was introduced in 2019 (which is considered old in the rapidly advancing field of computer vision. IKC is likely the first deep learning-based blind SR method). Although it seems outdated, I think the selection of IKC is quite appropriate. Firstly, I believe the SR of IMC images is different from the general SR research in computer vision. For natural images, visual quality is the priority. Therefore, many Blind SR methods have emerged post-IKC to enhance visual quality, such as Real-ESRGAN and BSRGAN. However, these come at the cost of objectivity in the resulting images. For the processing of IMC images proposed in this paper, these methods are not suitable. Instead, methods based on pre-defined degradation models are more conducive to ensuring the objectivity and authenticity of the resulting images. I didn't see relevant discussions in the article, but I believe these discussions are important.

From a computational standpoint, the method presented in this paper has lower innovation, but is reasonable and effective. It's hard for me to evaluate the application value as I am not familiar with IMC and its related industries and applications. I can see that the method improves the accuracy of subsequent detection and segmentation, but I am not quite sure about the real-world value of these numerical improvements.

We sincerely appreciate reviewer's time in reading and commenting on our manuscript. We have briefly summarized the Reviewer's inquiries in the following three aspects, and please do let us know if additional concerns remain unaddressed:

1. The denoising module may not work well for noise with spatial correlations.
2. There is no discussion that the method we used is appropriate and reasonable for IMC images.
3. Uncertainty about the practical value of the IMC image quality improvement.

Here are our responses to these three inquiries:

Response to Inquiry 1

We are very appreciative to the reviewer for pointing out the current limitations of our model. In

fact, as our background was more in analytical chemistry and proteomics, we did not take the spatial correlation noise into primary account when constructing our model. There are two main reasons for this: (1) most current studies in images denoising assume that the noise is sampled independently and do not consider the spatial correlation noise (*Nat Methods*, 2021, 18, 1395–1400, *Nat Methods*, 2023, 20, 1581–1592). (2) the distribution of proteins on IMC images is closely related to the spatial structure of the specimen, as shown in the **Figure 1** below. If spatial correlation noise is taken into account, it will be challenging to distinguish whether a pixel is a valid protein distribution or just noise. Fortunately, the experimental results show that our method is practically suitable for IMC images.

Figure 1 Protein distribution correlates with the spatial structure of the specimen

According to the reviewer's request, we have added more discussions of spatial correlation noise in the Supplementary Information.

Page 6, lines 135: Noise with spatial correlation is not taken into primary account in the network construction since existing studies generally assume that the noise is independently sampled^{9, 10}. And for IMC images, the distribution of proteins is generally correlated with the spatial structure of the specimen. If the noise has spatial correlations, the denoising module may encounter issues in distinguishing whether the single is a valid protein expression signal or simply just noise. Fortunately, our experimental results show that SpiDe-Sr is practically suitable for IMC images.

9 Eom, M., Han, S., Park, P. et al. Statistically unbiased prediction enables accurate denoising of voltage imaging data. *Nat Methods* 20, 1581–1592 (2023).

10 Li, X., Zhang, G., Wu, J. et al. Reinforcing neuron extraction and spike inference in calcium imaging using deep self-supervised denoising. *Nat Methods* 18, 1395–1400 (2021).

Response to Inquiry 2

We are very grateful to the reviewer for the comment that we should discuss whether our method is appropriate and reasonable for IMC images. We have thus discussed the differences between IMC images and natural images in the Introduction of the revised manuscript. We have also discussed in the Method session that the strategies we used were indeed appropriate and reasonable for IMC.

Page 2, lines 64: For natural images, visual quality is the priority. But for IMC images, rational enhancement is the necessary foundation for subsequent analysis. The IMC images enhanced by the existing unsupervised SR network lacks rationality due to the absence of ground truth^{19, 22}.

19 Qiao, C., Li, D., Liu, Y. et al. Rationalized deep learning super-resolution microscopy for sustained live imaging of rapid subcellular processes. *Nat Biotechnol* 41, 367–377 (2023).

22 Kim, J., Rustam, S., Mosquera, J.M. et al. Unsupervised discovery of tissue architecture in multiplexed imaging. *Nat Methods* 19, 1653–1661 (2022).

Page 22, lines 727: The rationality of the enhanced IMC image was the primary consideration when selecting the appropriate strategies. And the methods based on predefined degradation models were more conducive to ensure the objectivity and authenticity of the resulting images. Therefore, the idea of iteratively correcting the predefined blur kernel was opted in our study for super-resolution of IMC images after denoising.

Response to Inquiry 3

The significance of spatial proteomic research is to link proteomic data with spatial landscape and typological layout of cells, so as to gain insights into the spatial microenvironment of tissues and discover more precise biomarkers or new functional mechanisms (*Nature*, 2020, 578, 615 – 620; *Nature*, 2022, 601, 658-661; *Cell*, 2022, 185(2), 299-310).

There is a standardized data processing procedure for discovering new biomarkers from raw spatial proteomic data. This process includes image pre-processing, cell segmentation, cell protein expression calculation, cell clustering, and cell information analysis for practical applications. Each step in the process generates errors that accumulate as the process progresses. When it comes to the endpoint biological investigation analysis step, the accumulated errors largely determine the accuracy of the analysis conclusions. Some studies have proposed methods to make cell segmentation more accurate (*Nat Methods*, 2021, 18, 100-106) and some to make cell clustering more accurate (*Cell*, 2015, 2, 162(1):184-97, *Genome Biol*, 2019, 20, 297). Herein, what we have improved is the image pre-processing step by proposing the SpiDe-Sr to optimize the raw images, which also effectively reduces the accumulated errors and achieves more accurate application analysis.

As indicated in our practice, if IMC data from breast cancer patients were not processed by SpiDe-Sr, there was no indication that Gram-negative or Gram-positive bacteria had any particular correlation in the breast cancer microenvironment. The results were shown in Supplementary Fig. 8. But with the same raw data after SpiDe-Sr optimization (the methods in all steps remained the same), we found that the expression of Gram-negative bacterial marker was negatively correlated with the expression of immune cell marker, and the expression of Gram-positive bacterial marker was positively correlated with the immune cell marker. Similar conclusions were obtained in the analysis of label-free proteomics data, which cross-verified the accuracy of our results. The results of the analysis of the label-free proteomic data were presented in Supplementary Fig. 9 and Supplementary Fig. 10.

Obtaining spatial proteomic data from clinical cohorts is time-consuming and expensive, and it is

disappointing that new biological knowledge cannot be accurately discovered from the data. Thus, our method allows the clinical raw data to be analyzed more accurately, increasing the potential for discovering new biological insights, which we believe is the major practical value of our method.

Accordingly, we have added the results of data analysis without SpiDe-Sr enhancement in the revised manuscript.

Page 17, lines 556: Without SpiDe-Sr enhancement, B cells and T cells could not be distinguished and only 4 tumor cell clusters were identified based on the same IMC dataset because of noise interference or insufficiently precise details (Supplementary Fig. 8e-f). And in subsequent analyses, there was no indication that G- or G+ bacteria had any particular correlation in the breast cancer microenvironment (Supplementary Fig. 8i). After SpiDe-Sr enhancement, more biological information was mined.

Reviewer #2:

Chen et al. proposes a new algorithm, called SpiDe-Sr, for denoising of IMC images with the goal of improving the assay resolution. The SpiDe-Sr potentially addresses a very important limitation of IMC data analysis. However, the evaluation of manuscript in its current form is almost impossible due several reasons, some of which are noted here. I hope these comments would help the authors in revising their manuscript.

The major issue is that Che et al. fails to clearly communicate its findings, especially for broad readership of Nature Communications.

There are numerous grammatical mistakes throughout the manuscript. Several figure panels are not discussed (why are they include in the first place?)

We sincerely appreciate the reviewer for the time in reading and commenting on our manuscript. We do apologize for the unclarity in our previous manuscript. We have carefully studied each comment and revised our manuscript accordingly.

The presentation and grammar of the entire manuscript have been thoroughly revised. We have also employed native speakers to facilitate the grammatical proofread of the revised manuscript.

The figures, which the reviewer felt were not fully discussed, should have been the extended data figures. In previous manuscript, we placed these extended data figures (Extended data Fig1-Extended data Fig10) at the end of the manuscript. And we then uploaded Fig. 1 to Fig. 6 separately without figure legends because we were concerned that the figures imbedded in the manuscript were not clear enough after compression. This may have caused the figures to be misplaced in the final merged PDF file. We do apologize that this issue had made your reading disjointed. For this issue, we have removed the Extended data figures from the manuscript and placed them on page 14-33 in the Supplementary Information in the revised manuscript. We also provide detailed elaborations under each supplementary figure in the Supplementary Information. And all panels of figures in the revised manuscript were also discussed.

In summary, it is utterly impossible to evaluate the scientific merit of the manuscript given its poor presentation. Some of the sentences are incomprehensible. There are many unnecessary abbreviations (do the authors really need to abbreviate “state-of-the-art” to SOTA?). On the other hand, there are abbreviations with unclear definition (e.g. LA, LB, TNBC, etc.)

We sincerely appreciate your criticism. We have employed native speaker to revise the grammar and diction throughout the manuscript. And the presentation of the entire manuscript was also carefully revised.

In the process of completing the manuscript, we have carefully referenced high-quality articles. We noticed that “State-of-the-art” was abbreviated as SOTA in previous references in our field (*Nat Biotechnol*, 2023, 41, 367–377, *Nat Mach Intell*, 2021, 3, 581–589). In the revised manuscript, we have also carefully checked all the abbreviations to ensure they were defined in the text or in the figure legend when they were first used. We clarified the definition of the abbreviation (HER2, LA, LB, TNBC) in the figure legend of Fig. 5 in the manuscript according to the general denotation in tumor research community.

Page 16, lines 501: HER2, human epidermal growth factor receptor 2 breast cancer. LA, luminal A breast cancer. LB, luminal B breast cancer. TNBC, triple negative breast cancer.

Considering the general readers of *Nature Communications* and the reviewer’s comment, we have avoided the use of unnecessary abbreviations in the main text of revised manuscript.

Performance metrics are not defined. What is SSIM? What is pSNR? How “accuracy of cell extraction” is defined / calculated Definition of terminologies/ metrics / etc should be clarified, especially for a journal such as Nature Communications with a broader readership. A short introduction of methods included in comparative analysis, tools used in throughout the paper, metrics used for evaluation, etc. would help potential readers (as well as reviewers). This is not an IEEE journal.

We greatly appreciate the detailed comments from the reviewer. According to the reviewer’s request, we have clarified the definitions of the abbreviations PSNR, SSIM and other metrics in the manuscript where they were used for the first time.

Page 4, lines 107: **Abbreviations and remarks:** PSNR, peak signal-to-noise ratio, larger means less noise. SSIM, structural similarity, larger means more similar to the ground truth. SOTA, state-of-the-art.

Page 4, lines 138: The peak signal-to-noise ratio (PSNR) and structural similarity (SSIM) were calculated between the ground truth and the blurred images before and after SpiDe-Sr enhancement (Details are provided in the **Methods** section).

Page 8, lines 238: Therefore, intersection over union score (IoU) was calculated to evaluate the accuracy of cell segmentation⁰.^{Error! Reference source not found.}

20 Li, X., Zhang, G., Wu, J. et al. Reinforcing neuron extraction and spike inference in calcium imaging using deep self-supervised denoising. *Nat Methods* 18, 1395–1400 (2021).

32 Jia Y., Yuning J., Zhangyang W., et al. UnitBox: An Advanced Object Detection Network. In Proceedings of the 24th ACM international conference on Multimedia (MM 16). Association for Computing Machinery, New York, NY, USA, 516–520 (2016).

Page 16, lines 499: **Abbreviations:** CH, Calinski-Harabasz score; DB, Davies-Bouldin score; *t*-SNE, t-distributed stochastic neighbor embedding; G⁻, gram-negative bacteria; G⁺, gram-positive bacteria; HER2, human epidermal growth factor receptor 2 breast cancer; LA, luminal A breast cancer; LB, luminal B breast cancer; TNBC, triple negative breast cancer; ER, estrogen receptor; PR, progesterone receptor.

Page 17, lines 525: Calinski Harabasz (CH) score⁰, which evaluated the degree of dispersion between clusters, was increased by 38.29± 24.23%, indicating the identified clusters were more discrete. Meanwhile, the Davies-Bouldin (DB) score⁰, which evaluated the intra-cluster tightness, was reduced by 11.12± 8.73%, indicating more similarity within the identified clusters (**Fig. 5e**).

38 Calinski, T., and Harabasz, J. A Dendrite Method for Cluster Analysis, Communications in Statistics, 1974, 3, 1-27.

39 William, H.E. Day, Validity of clusters formed by graph-theoretic cluster methods. Mathematical Biosciences, Volume 36, Issues 3–4, 1977, 299-317, ISSN 0025-5564.

Meanwhile, we have also described in details the definition of each performance metrics and how it was calculated in the Performance Metrics section in the manuscript.

Page 26, lines 900~ Page 28, lines 958.

Here are a non-exhaustive list of examples that hopefully could guide authors in their revision. What do authors mean by “the same underlying scene were required”, given imaging the same section used for IMC is impossible? How “same” is determined?

We sincerely appreciate the reviewer’s inquiry. In the phrase "the same underlying scene", we have followed the wording in the reference (*Nat Methods, 2021, 18, 1395–1400*). For instance, as shown in **Figure. 1** below, **a** is considered a clean image, which is also usually considered to be ground truth. **b** is generated by **a** overlaid with Gaussian noise and **c** is generated by **a** overlaid with pepper noise. We consider **b** and **c** to have the same underlying scene. Gaussian noise and pepper noise simulate noises that may be generated during different imaging processes, respectively.

Figure. 1 **a.** The clean image, which is considered ground truth in the three images. **b.** Image in **a.** added with Gaussian noise. **c.** Image in **a.** added with pepper noise.

When photographing a field of view under a microscope, different focuses or different lighting will result in different images. Images taken at different focuses or under different lighting are

considered to have the same underlying scene because essentially the biological content in images is the same. The imaging principle of IMC is that the laser ablates the sample and then mass spectrometry detects the products of the ablation to identify the proteins in the sample. Different from the microscope that can repeatedly photograph the same sample, for IMC, the same sample can only be ablated once, and the instrument can only image the same sample once. Thus, obtaining multiple IMC images with the same underlying scene is practically not feasible.

Figures are unclear. The legends are missing. Considering Figure 3 as an example, what is GT? What do red / gray legends in panel i represent? In the same panel, what are those numbers? This is just an example. Similar problems exist throughout the manuscript.

We appreciate the comments from the reviewer. We apologize for unclear figures and disappearing legends caused by irregularities in our original manuscript. We have made the details clearer in the legends for each figure in the manuscript, including an overview of each small panel, what the different colors represent, the abbreviations involved, and the notes.

Page 3, lines 92: **Fig. 1 SpiDe-Sr method.** **a**, The architecture of SpiDe-Sr. The network was comprised of the denoising module and the super-resolution module. The denoising module included the neighbor sub-sampler and the U-net denoising network. And the super-resolution module had three components: the blur kernel predictor (P_θ), the blur kernel corrector (C_θ) and the image super-resolution network (S_θ). **b**, Inference using the trained SpiDe-Sr network. The architectural details and interpretability of the SpiDe-Sr were illustrated in Supplementary **Fig. 1** and Supplementary **Fig. 2**. **c**, Quantitative evaluation of SR image quality with different iterations of blur kernel estimation. Dashed line indicated the optimal number of iterations. $n=4392$ images. **d**, Quantitative evaluation of image PSNR and SSIM with different noise levels before and after the SpiDe-Sr enhancement, $n=4392$ images. **e**, The number of cells extracted based on images with different noise levels before and after the SpiDe-Sr enhancement. Total number of cells in the field of view is 200. **f**, Visual comparison of SpiDe-Sr method with the three SOTA super-resolution methods including SRCNN, KernelGAN, and RCAN. **g**, Spatial profiles of extracted cells in the field of View 1. Correctly segmented regions (true positives) were colored in green. Missing (false negatives) and extra regions (false positives) were colored in red and gray, respectively. All cell segmentation tasks in our work were implemented with the Cellpose algorithm. **Abbreviations and remarks:** PSNR, peak signal-to-noise ratio, larger means less noise. SSIM, structural similarity, larger means more similar to the ground truth. SOTA, state-of-the-art.

Page 6, lines 185: **Fig. 2 Validation of SpiDe-Sr on IMC images of MCF-7 cell line.** **a**, Schematic of acquiring paired images of cells with fluorescent/ metal dual-labeled antibodies. **b, c, d**, Confocal microscopy (left), raw IMC (middle), and SpiDe-Sr enhanced IMC (right) images of nucleus and examples of relatively high/ moderate/low expression markers (**b**, Tubulin/ **c**, CD45/ **d**, CD34). Cell segmentation was conducted with Cellpose. Missed (false negatives) and extra segmentations (false positives) were respectively indicated by yellow and green arrows. Correctly extracted but wrongly bounded regions were indicated by red arrows. **e- f**, Violin-scatter plots showing the distribution of (**e**) peak signal-to-noise ratio (PSNR) and (**f**) structural similarity (SSIM) with ground truth (GT) images before and after SpiDe-Sr enhancement. Each gray line

represented the variation of a single image before and after enhancement. n=52 (Tubulin)/ 36 (CD45)/ 71 (CD34) images. **g**, Accuracy of cell extraction before and after SpiDe-Sr enhancement. **h**, Violin-scatter plots showed the distribution of intersection over union (IoU) of accurately extracted cells in IMC images before and after SpiDe-Sr enhancement vs. GT images. Each line represented the variation of a single cell before and after enhancement. Increasing and decreasing pairs were colored in gray and red, respectively. **i**, Violin-scatter plots showed the distance of biomarker expressions in accurately extracted cells from IMC images with and without SpiDe-Sr enhancement to the corresponding cells in GT images. Each line represented the variation of a single cell before and after enhancement. Increasing and decreasing pairs were colored red and gray, respectively. **j**, Normalized marker expressions in accurately extracted cells. In **h-j**, the number of cells was 216/ 241/ 357. **k**, Comparison of SpiDe-Sr method with the three competitive SR methods in PSNR, SSIM, and running time. **l**, Visual comparison of SpiDe-Sr method with the three competitive super-resolution methods. In **e-f**, asterisks indicated statistical significance by paired-samples *t*-test, ** $P < 0.01$, *** $P < 0.001$.

Page 9, lines 280: **Fig. 3 Validation of SpiDe-Sr on IMC images of mouse fatty liver tissues.** **a**, Schematic of acquiring paired images of mouse fatty liver tissues with fluorescent/metal dual-labeled antibodies. **b, c, d**, Confocal microscopy (left), raw IMC (middle), and SpiDe-Sr enhanced IMC (right) images of nucleus and examples of relatively high/moderate/low expression markers (**b**, Tubulin/ **c**, CD45/ **d**, CD34). Cell segmentation was conducted with Cellpose. Missed (false negatives) and extra segmentations (false positives) were respectively indicated by yellow and green arrows. Correctly extracted but wrongly bounded regions were indicated by red arrows. **e-f**, Violin-scatter plots showing the distribution of (**e**) peak signal-to-noise ratio (PSNR) and (**f**) structural similarity (SSIM) with ground truth (GT) images before and after SpiDe-Sr enhancement. Each gray line represented the variation of a single image before and after enhancement. n= 37 (Tubulin)/ 37 (CD45) / 21 (CD34) images. **g**, Accuracy of cell extraction before and after SpiDe-Sr enhancement. **h**, Violin-scatter plots showed the distribution of intersection over union (IoU) of accurately extracted cells in IMC images before and after SpiDe-Sr enhancement vs. GT images. Each line represented the variation of a single cell before and after enhancement. Increasing and decreasing pairs were colored in gray and red, respectively. n= 235 (Tubulin)/ 422 (CD45)/ 203 (CD34) cells. **i**, Violin-scatter plots showed the distance of biomarker expressions in accurately extracted cells from IMC images with and without SpiDe-Sr enhancement to the corresponding cells in GT images. Each line represented the variation of a single cell before and after enhancement. Increasing and decreasing pairs were colored red and gray, respectively. **j**, Normalized marker expressions in accurately extracted cells. **k**, Comparison of SpiDe-Sr method with the three competitive SR methods in PSNR, SSIM, and running time. **l**, Visual comparison of SpiDe-Sr method with the three competitive super-resolution methods. In **e-f**, asterisks indicated statistical significance by paired-samples *t*-test, *** $P < 0.001$.

Page 12, lines 375: **Fig. 4 Validation of SpiDe-Sr on IMC images of human breast cancer tissues.** **a**, Schematic of acquiring paired images of breast cancer tissues with fluorescent/metal dual-labeled antibodies. **b, c, d**, Confocal microscopy (left), raw IMC (middle), and SpiDe-Sr enhanced IMC (right) images of nucleus and examples of relatively high/moderate/low expression markers (**b**, Tubulin/ **c**, CD45/ **d**, CD34). Cell segmentation was conducted with Cellpose. The

missed (false negatives) and extra segmentations (false positives) were respectively indicated by yellow and green arrows. Correctly extracted but wrongly bounded regions were indicated by red arrows. **e-f**, Violin-scatter plots showing the distribution of (**e**) peak signal-to-noise ratio (PSNR) and (**f**) structural similarity (SSIM) with ground truth (GT) images before and after SpiDe-Sr enhancement. Each gray line represented the variation of a single image before and after enhancement. $n=47$ (Tubulin)/ 25 (CD45)/ 54 (CD34) images. **g**, Accuracy of cell extraction before and after SpiDe-Sr enhancement. **h**, Violin-scatter plots showed the distribution of intersection over union (IoU) of accurately extracted cells in IMC images before and after SpiDe-Sr enhancement vs. GT images. Each line represented the variation of a single cell before and after enhancement. Increasing and decreasing pairs were colored in gray and red, respectively. **i**, Violin-scatter plots showed the distance of marker expressions in accurately extracted cells from IMC images with and without SpiDe-Sr enhancement to the corresponding cells in GT images. Each line represented the variation of a single cell before and after enhancement. Increasing and decreasing pairs were colored red and gray, respectively. **j**, Normalized marker expressions in accurately extracted cells. In **h-j**, $n=244$ (Tubulin)/ 207 (CD45)/ 240 cells. **k**, Comparison of SpiDe-Sr method with the three competitive SR methods in PSNR, SSIM, and running time. **l**, Visual comparison of SpiDe-Sr method with the three competitive super-resolution methods. In **e-f**, asterisks indicated statistical significance by paired-samples *t*-test, $*P < 0.05$, $**P < 0.01$, $***P < 0.001$.

Page 15, lines 470: **Fig. 5 Application of SpiDe-Sr to spatial proteomics data from four major subtypes of breast cancer patients.** **a**, Workflow of IMC image acquisition. **b**, A raw breast cancer IMC image and cells extracted by Cellpose based on it (left), compared with the corresponding SpiDe-Sr enhanced image and cells extracted based on it (right). **c**, Number of cells extracted in all breast cancer samples. **d**, Normalized expressions of 14 markers in four breast cancer subtypes at single-cell level ($n=8,697/5,550/7,316/5,393$ for HER2/ LA/ LB/ TNBC). **e**, Comparison of CH and DB scores of FlowSOM clustering results of cells extracted from all acquired images with and without SpiDe-Sr enhancement. The clustering was repeated 10 times for each cluster number interval. **f**, Comparison of CH and DB of PhenoGraph clustering results of cells extracted from all acquired images with and without SpiDe-Sr enhancement. The clustering was repeated 120 times without preset cluster number. Histogram showed the frequency distribution of output cluster numbers. In **e-f**, both CH and DB were statistically different before and after SpiDe-Sr enhancement (*t*-test, $P < 0.001$). **g**, The clustering results with the highest CH score. The heatmap (left) showed normalized mean marker expressions of each PhenoGraph cluster. The stacked bar plot showed (middle) the proportions of four subtypes of breast tumor cells in each cluster. The bar plot (right) showed the absolute cell counts in each cluster. **h**, *t*-SNE map of 269,556 cells sub-sampled from all acquired images. Cell types were manually identified and marked by different colors. **i-j**, The proportion of cells of each breast cancer subtype in the clusters which had the highest expression of G^- (**i**, C12, $n=144/506/76/121$ cells) and G^+ (**j**, C10, $n=21/78/3,654/101$ cells) bacterial markers, compared to the total cell count of each subtype. **k**, (below) Heat map showing the Pearson correlation coefficients of the 14 markers in C12 with each other. (above) Heat map showing the Pearson correlation coefficients of the 14 markers in C10 with each other. Positively and negatively correlated markers were colored in orange and blue, respectively. LPS and LTA were colored in red. G^- bacteria were universally positive correlated to

tumor markers (ISG15 and IFI6) and negative correlated to immune markers (CD45 and CD68) in all four subtypes of breast cancers, while G^- bacteria showed completely reversed patterns. **l-m**, Box plots showed the absolute expressions of 14 markers in C12 (**l**, $n=847$ cells) and C10 (**m**, $n=3,854$ cells) of the four breast cancer subtypes. Red and blue asterisks respectively represented the statistical significance of proteins positively and negatively associated with LPS/ LTA versus LPS/ LTA (t -test, $**P < 0.01$, $***P < 0.001$). **Abbreviations:** CH, Calinski-Harabasz score; DB, Davies-Bouldin score; t -SNE, t -distributed stochastic neighbor embedding; G^- , gram-negative bacteria; G^+ , gram-positive bacteria; HER2, human epidermal growth factor receptor 2 breast cancer; LA, luminal A breast cancer; LB, luminal B breast cancer; TNBC, triple negative breast cancer; ER, estrogen receptor; PR, progesterone receptor.

Page 18, lines 565: **Fig. 6 Migrating SpiDe-Sr to fluorescence microscopy images. a**, Paired images at different magnifications (10x and 40x) was acquired for MCF-7 cell line, mouse retina, and breast tissue. **b-d**, Raw images at 10x magnification (left) of MCF-7 cell (**b**), Mouse retina (**c**), and breast tissue (**d**), and corresponding 40x images reconstructed from the 10x images using SpiDe-Sr (middle), along with the true blur kernels and the blur kernels between the 10x and SpiDe-Sr enhanced 40x images (right). **e**, Comparison of the super resolution images reconstructed by SRCNN, KernelGAN, RCAN and SpiDe-Sr for the three sample types. **f-g**, Comparisons of PSNR (**f**) and SSIM (**g**) among the four super resolution methods in different sample types. $n=65/16/22$ for MCF-7 cells/ mouse retina tissues/ human breast tissues. **h**, Overall comparison of the PSNR, SSIM, and running time among the four super resolution methods. **i**, Visual comparison of 40x ground truth image of F-actin and 40x super-resolution image reconstructed from 10x image using SpiDe-Sr, as well as the other three methods. **Abbreviations:** FFPE, formalin fixed paraffin embedded.

What is the difference between “penetrating” and “existing” tumor microenvironment studies? By the way, what is the meaning of penetrating analysis?

We sincerely appreciate the reviewer’s inquiry and apologize for the unclear wording choices in the manuscript. We realize the use of “penetrating” does not deliver accurate meaning to readers and reviewers, thus we have changed all “penetrating” to “precise” in the revised manuscript:

Page 16, lines 505: **SpiDe-Sr facilitates precise spatial proteomics analysis of breast cancer microenvironment**

Page 16, lines 509: Therefore, SpiDe-Sr was adopted to enhance the multiplex IMC images for higher resolution so that bacterial signals could be precisely analyzed.

For IMC data analysis in mass spectrometry community, there is a recognized standard process (*Nature*, 2020, 578, 615–620) that includes image pre-processing, cell segmentation, cell protein expression calculation, cell clustering, and practical application analysis. Each step in the process generates errors that accumulate as the process progresses. The accumulated error impacts the accuracy of the actual biological application analysis. Some studies have proposed methods to reduce the errors in cell segmentation (*Nat Methods*, 2021, 18, 100-106) and clustering (*Genome Biol*, 2019 20, 297), and our method is to directly optimize the raw image so that the errors in the subsequent steps can be reduced as a whole.

As indicated in your clinical application, if IMC data from breast cancer patients were not processed by SpiDe-Sr, there was no indication that Gram-negative or Gram-positive bacteria had any particular correlation in the breast cancer microenvironment. The analysis results of the data without SpiDe-Sr enhancement under the standard process were shown in Supplementary Fig. 8. But when the same data was processed by SpiDe-Sr and then analyzed under the standard process (the methods in all steps remained the same), we found that the expression of Gram-negative bacterial marker was negatively correlated with the expression of immune cell marker, and the expression of Gram-positive bacterial marker was positively correlated with the immune cell marker. Similar conclusions were obtained in the analysis of label-free proteomic data, which cross-verified the accuracy of our analyzed results. The results of the analysis of the label-free proteomic data were presented in Supplementary Fig. 9 and Supplementary Fig. 10. Thus, after using our method, more biological information was indeed mined.

Line 588: what do authors mean by “8 diverse tumor cell phenotype”? How do they know the phenotype of cells?

We truly appreciate the reviewer’s inquiry. We followed the definition of “phenotype” in reference (*Nature, 2020, 578, 615–620*), referring to the type and expression of proteins in a cell in the specific environment. Taking into account the reviewer’s comments and in order to make the manuscript comprehensible to the reader, we have replaced the terminology "cell phenotype" with "cell cluster" in the manuscript.

Page 17, lines 534: Normal healthy cells (C1- C9), B cells (C32 and C33 with highest expression of CD19), T cells (C26 with highest expression of CD45), macrophage (C11 with highest expression of CD68), and cells containing G⁻/ G⁺ bacteria (C12/ C10), as well as 8 diverse tumor cell clusters were identified clearly (**Fig. 5g-h**).

Page 20, lines 630: We focused on G⁻ and G⁺ bacteria relevant tumor cell cluster.

We obtained the expression of 14 proteins in 269,556 cells after cell segmentation. Cells with similar protein expression were divided into a cluster based on the expression of these 14 proteins. As shown in **Fig. 5g**, all cells were divided into 33 cell clusters based on the expression of proteins. The redder the color in the heat map, the higher the protein expression.

CD19 is highly expressed in B cells and is a marker for B cells, therefore we classified the two clusters C32 and C33, which highly express CD19, as B cell clusters. Similarly, we classified the C26/ C11/ C27 cluster with high expression of CD45/ CD68/ CD8a as a T cell/ Macrophages/ CD8+T cells cluster, respectively.

ER, HER2, IFI6, ISG15, Ki67, PKCD, PR and ZC3HVI are common markers of breast cancer according to WHO website and references (*Nature, 2020, 578, 615 - 620*). As shown in **Fig. 5g**, the C16 cluster with high expression of ER was classified to be an ER⁺ cell cluster, a tumor cell cluster. C18 with high expression of HER2 is classified as a HER2⁺ tumor cell cluster. Using the same classification pipeline, 6 other tumor phenotype clusters were distinguished, including ISG15⁺, Ki67⁺, PR⁺, PKCD⁺, IFI6+ISG15⁺ and PKCD+ZC3HV1⁺. If biomarkers of neither immunity nor tumor were expressed, as in the case of C1 to C9, they were considered normal healthy cells. As shown in the **Figure 2**, starting from the protein columns, we identified the

clusters with the highest biomarker expressions and determined the corresponding cell phenotype clusters.

Figure 2 Determining cell clusters from heat map.

IFI6 is just a marker of apoptosis. What do the author mean my stating that “correlated with tumor makers, especially IFI6” (Line 596)?

We sincerely appreciate the reviewer’s inquiry. We agree with the reviewer that, on its own, IFI6 is really just a marker of apoptosis. However, since IFI6 is abundantly expressed in many cancer tissues, it is also considered as a tumor marker in many references (*Nat Microbiol*, 2018, 3, 1214-1223, *Cell Dev Biol*, 2021, 25;9:677697, *Br J Cancer*, 2018, 119, 52-64).

And in our data from the breast cancer microenvironment, we found that the expression of gram-negative bacterial marker (LPS) was positively correlated with IFI6, indicating that cells with high LPS expression also had high IFI6 expression. As shown in the **Figure. 3** below, positive correlations were in red and negative correlations were in blue. Pearson correlation coefficients greater than 0.75 were marked with two asterisks, while Pearson correlation coefficients greater than 0.5 were marked with one asterisk.

Figure. 3 Correlation between LPS expression and other biomarker expressions.

The conclusions from the above figure were summarized in the following **Table 1**.

Table 1 Correlation between LPS expression and other biomarker expressions

Sub-types of breast cancer	Positively associated with LPS ($P > 0.5$)	Negatively associated with LPS ($P > 0.5$)
HER2	ISG15, IFI6 , HER2, ZC3HV1	CD45, CD68
LA	IFI6 , PKCD, CD19, ZC3HV1, ISG15	\
LB	IFI6	CD68
TNBC	CD19, CD45, PKCD, ZC3HV1, ISG15	CD8a

From the **Table 1**, we found that the expression of LPS was positively correlated with tumor markers, such like IFI6.

According to the reviewer's inquiry, we have also added the Table 1 to the Supplementary Information at Page 43, lines 646. **Supplementary Table 8 Correlation between LPS expression and other biomarker expressions**

What do the authors mean by "associated with malignant appreciation, namely HER2 and Ki67" (Lines 600-601). How malignancy can be appreciated?

We sincerely appreciate the reviewer's inquiry. Based on reviewer's inquiry, we realize that "malignant appreciation" is an inaccurate expression, and we apologize for this misleading terminology. We have changed the wording in the revised manuscript.

Page 17, lines 543: Inversely, in C10, the expression of immune markers, such as CD45, was positively correlated with LTA expression in all four breast cancer subtypes except LB, and the expression of LTA was negatively correlated with the expression of breast cancer markers associated with abnormal cell growth, namely HER2 and Ki67 (upper half of Fig. 5k, Supplementary Fig. 7e and Table 9).

Similar to our responses to the last question, the heat maps of Pearson correlation coefficient between LTA expression and other biomarker expressions are shown in Figure 4. And we have summarized the Figure 4 in the Table 2 below.

Figure. 4 Correlation between LTA expression and other biomarker expressions.

Table 2 Correlation between LTA expression and other biomarker expressions

Sub-types of breast cancer	Positively associated with LTA ($P > 0.75$)	Negatively associated with LTA ($P > 0.75$)
HER2	CD45, PR, CD8a, Ki67	HER2, LPS
LA	CD68, LPS, IFI6, ZC3HV1	Ki67, PR
LB	LPS	\
TNBC	ER, ISG15, PR, CD45, IFI6	\

Because both HER2 and Ki67 are associated with abnormal cell growth, we thus identify that LTA may be associated with abnormal cell growth based on the Table 2.

Per the reviewer's inquiry, we have also added the **Table 2** to the Supplementary Information at Page 44, lines 680. **Supplementary Table 9 Correlation between LTA expression and other biomarker expressions**

The authors stated that "The most straightforward strategy to obtain IMC images with high PSNR is to manually eliminated pixel values above and below empirical thresholds". How are empirical thresholds chosen? A brief explanation is needed.

We sincerely appreciate the reviewer's inquiry. Manual elimination of pixel values above and below empirical thresholds is usually done in MCD Viewer, the software that accompanies IMC instruments. Specific instructions are available on the Fluidigm instrument's official website (<https://www.standardbio.com/products/technologies/imaging-mass-cytometry>). To do so, we import the original file into the MCD Viewer, then select the channel of interest, and adjust the two parameters of Threshold Min and Threshold Max so that there is no background noise on the image and the cell nuclei are clear.

As shown in **Figure 5**, take the cell nucleus channel (DNA_191Ir) image as an example. If the background noise is too high, IMC users would adjust the parameter Threshold Min to remove the pixels below the threshold. If the cell nucleus is too bright, resulting in unclear cell boundaries or too many discrete bright spots, the Threshold Max can be adjusted to remove pixels above the threshold. In our experience, the Threshold Min is generally adjusted between 0.5- 5, while the adjustment range of Threshold Max will be larger, generally according to the specific channel. Adjustment panel is shown in **Figure 5.a**, where the image before adjustment is **b**, the image after adjustment is **c**. In **Figure 5.c**, there is less noise and the cell boundaries are clearer than **b**.

Figure 5 a, Operation interface. **b**, Image with low PSNR, **c**, Image with high PSNR.

At line #216, the authors stated that “In addition, SpiDe-Sr was visually superior to three state-of-the-art (SOTA) single image SR methods including SRCNN, KernelGAN and RCAN”. Quantitative superiority should be established by showing results from "SOTA" SR methods in Figures 1c, d, e and all other figures, where breast tumors and mouse samples are discussed.

We appreciate the reviewer’s suggestion. IMC samples archived in our laboratory, including 91 breast cancer samples, 67 liver cancer samples, and 63 mouse samples were cropped to obtain a total of 21,960 raw images. In order to obtain the maximum amount of data, we did not differentiate the images according to the sample type, but rather mixed all the images and divided them into the 1) training set, 2) the validation set, and 3) the test set for subsequent model construction. All three datasets contained breast cancer samples, liver cancer samples, and mouse samples, as presumed. The parameters of the model were determined based on the results obtained from the overall validation set. The results in **Fig. 1c** and **d** were obtained on the test set. All three types of samples were involved together in the construction and evaluation of the model. Hence, the main purpose of **Fig. 1** was to illustrate the structure and benchmark performance of the model.

According to the reviewer’s inquiry, we have further described the use of the images in the manuscript for better clarity.

Page 21, lines 691: The IMC samples archived in our laboratory (including 91 breast cancer samples, 67 liver cancer samples and 63 mouse organs samples) were prepared into 21,960 raw images of 300×300 pixels in TIFF format using MATLAB scripts. These 21,960 raw images constituted a dataset named SpiSet. Three-fifths of images in SpiSet were allocated for training the denoising network, and one-fifth were employed for validation. The remaining one-fifth of SpiSet

were randomly superimposed with Gaussian or Poisson or pepper noise through the utilization of built-in function within MATLAB for testing.

Where did the authors get the Ground truth of cell segmentation? In other words, how exactly is it known that which cell "extraction" (which the reviewer is unsure of its meaning) is correct/missing/extra when the sample is analyzed by Cellpose?

We truly appreciate the detailed reading and questions. We used the fluorescent/metal-dual labeling approach to obtain both IMC images, and confocal images with the same underlying scene as the IMC images. Because confocal image had higher PSNR and resolution, we thus used confocal image as the ground truth to evaluate performance. The confocal images were fed into the Cellpose program running with default parameters to obtain preliminary cell segmentation results. The default parameter settings according to the literature (*Nat Methods, 2021, 18, 1395–1400*) were shown in the **Table 3**. Then the expertise researcher adjusted Cellpose program parameters to obtain the accurate segmentation results from confocal images as the ground truth of cell segmentation. There are two parameters that should be adjusted, namely cell diameter (pixels) and model zoo. When the raw IMC and SpiDe-Sr enhanced IMC were fed into the Cellpose program, the program was only run with the default parameters, and the parameters were no longer manually adjusted.

Table 3 Default parameters for Cellpose

Parameters	Settings
Up/down or W/S	RGB
page up/down	image
brush size	3
MASK ON	Yes
single stroke	Yes
outlines on	No
scale disk on	Yes
use GPU	Yes
flow_threshold	0.7
cellprob_threshold	0.0
stitch_threshold	0.0
cell diameter (pixels)	calibrate
model zoo	default

It should be noted that the output of Cellpose was only the mask of cell segmentation of image. For the calculation of the accuracy of the cell extraction, we referred to the literature (*Nat Methods, 2021, 18, 1395–1400*). Whether the cells were accurately extracted, missed or extra detected, should be manually counted by the researcher. In **Figure 6**, we explained this pipeline with an example.

Figure 6 Example of the cell missing and extra. **a**, Example of the missing cell. **b**, Example of the extra cell.

The cell boundaries in red (**a**) / green (**b**) color in **Figure 6** were drawn from the cell segmentation mask output by Cellpose. There were 4 cells in the confocal that were extracted in **a**. There should be 4 cells in the raw IMC as well, but none of them were extracted in Cellpose, so before SpiDe-Sr, all 4 cells were missed. After SpiDe-Sr enhancement, all 4 cells were accurately detected in the image. In another example, there were 2 cells in the confocal in **b**, but 3 cells were extracted in the raw IMC image, so the cell \square was considered to be extra.

According to the reviewer's inquiry, we have also added a discussion of the ground truth for cell segmentation and **Table 3** in the Supplementary Information.

Page 13, lines 319: The confocal images were fed into the Cellpose program running with default parameters to obtain preliminary cell segmentation results. The default parameter settings according to the literature¹¹ were shown in the Supplementary **Table 10**. Then the expertise researcher adjusted the Cellpose program parameters to obtain the accurate segmentation results as the ground truth of cell segmentation. When the raw IMC and SpiDe-Sr enhanced IMC were fed into the Cellpose program, for the control variable, the program was only run with the default parameters, and the parameters were no longer manually adjusted. It should be noted that the output of Cellpose was only the mask of cell segmentation of image. And whether the cells were accurately extracted, or missed or extra extracted, should be manually counted by the researcher.

11 Stringer, C., Wang, T., Michaelos, M. et al. Cellpose: a generalist algorithm for cellular segmentation. Nat Methods 18, 100–106 (2021).

Page 45, lines 716: **Supplementary Table 10 Default parameters for Cellpose.**

Parameters	Settings
Up/down or W/S	RGB
page up/down	image
brush size	3
MASK ON	Yes
single stroke	Yes
outlines on	No
scale disk on	Yes
use GPU	Yes
flow_threshold	0.7
cellprob_threshold	0.0
stitch_threshold	0.0
cell diameter (pixels)	calibrate
model zoo	default

How does SpiDe-Sr compare with original images which were down-sampled (lines 207-208).

We appreciate the inquiry. In the phase of testing model benchmark performance, the raw IMC images were served as ground truth because of the lack of clean and high-resolution images. The raw images were superimposed with noise and down-sampled with bicubic interpolation to one-fourth of the original size to form blurred images. The blurred images were fed into SpiDe-Sr, and the images were denoised and enlarged by a factor of 4. At this stage, the images after SpiDe-Sr were as large as the raw images. Then we can calculate the PSNR and SSIM of enhanced images between the raw mages and the SpiDe-Sr enhanced images. The PSNR and SSIM of blurred images were calculated between the raw images and the blurred images after bicubic interpolation up-sampling.

According to the reviewer's inquiry, we have also added these details to the revised manuscript.

Page 4, lines 135: To quantitatively evaluate the benchmark performance of SpiDe-Sr, the raw IMC images were served as ground truth because of the lack of clean and high resolution images.

The raw images were superimposed with noise and down-sampled to one-fourth of the original size to form blurred images. The peak signal-to-noise ratio (PSNR) and structural similarity (SSIM) were calculated between the ground truth and the blurred images before and after SpiDe-Sr enhancement (Details are provided in the **Methods** section).

Page 23, lines 758: For testing or validation, bicubic interpolation was used to align the image sizes when the SpiDe-Sr was not required.

Many details of methods and procedures are missing. For instance, what parameters were used for Cellpose. Did the authors attempt to train a model for Cellpose (which is obviously possible and should be done). This would be a more accurate comparison.

We truly appreciate the reviewer's inquiry and comments. We apologize for the unclear details of methods and procedures in the original manuscript. We used Cellpose V1.0 (*Nat Methods* 2021, 18, 100-106), which cannot be retrained on the existing model to get specific parameters.

In the SpiDe-Sr performance validation phase, we used Cellpose for cell segmentation of confocal

and IMC images. The confocal images were input into the Cellpose program and run with default parameters (in Table 3) to obtain preliminary cell segmentation results. The two parameters, namely cell diameter (pixels) and model zoo, were then adjusted. The calibrated cell diameter was a numerical value that could be manually adjusted. The model zoo was set to cytoplasm pattern (cyto). As for IMC images, in order to avoid artificial bias, we did not manually correct these parameters again.

In addition, in the stage of breast cancer microenvironment analysis, we first used the default parameters for cell segmentation of breast cancer images. The model zoo was set to cytoplasm pattern (cyto). And then we adjusted the value of approximate cell diameter one by one in the user interface to make the segmentation as accurate as possible.

According to the reviewer's inquiry, we have added the setting of the Cellpose parameter to the manuscript:

Page 25, line 857: **Step 4** The regions of individual cells in all images were segmented at the pixel level using cytoplasm pattern with adaptive calibration diameter in Cellpose to generate masks. Other default parameters were shown in **Supplementary Table 10**. The mask for single-cell segmentation in each ROI was manually adjusted and selected.

Page 27, line 918: Cell extraction was regarded as an instance segmentation problem, accuracy and object-level metrics (IoU and F_1) were adopted to evaluate the segmentation performance of Cellpose before and after enhancement. Further details were in **Supplementary Note 7** and **Table 10**.

Also, we have added the details of the parameter tuning to the supplementary information.

Page 13, line 329: Specifically, in the SpiDe-Sr performance validation phase, we used Cellpose for cell segmentation of confocal and IMC images. The confocal images were input into the Cellpose program and run with default parameters (in **Supplementary Table 10**) to obtain preliminary cell segmentation results. The two parameters, namely cell diameter (pixels) and model zoo, were then adjusted. The calibrated cell diameter was a numerical value that could be manually adjusted for specific conditions. The model zoo was set to cytoplasm pattern (cyto). As for IMC images, in order to avoid artificial bias as much as possible, we did not manually correct these 2 parameters again. In the stage of breast cancer microenvironment analysis, we first used the default parameters for cell segmentation of breast cancer images. The model zoo was set to cytoplasm pattern (cyto). And then we readjusted the value of approximate cell diameter one by one in the user interface to make the segmentation as accurate as possible.

In addition to this, we have tried the trainable segmentation method CellProfiler (*Nat Methods*, 2012, 9(7), 714-716), which could be retrained to characteristically calibrate the parameters. The cell segmentation results of CellProfiler were not superior to those of Cellpose with default parameters. As shown in **Figure 7**, round nuclei could also be well segmented by CellProfiler, but for shuttle-shaped nuclei, the segmentation result of CellProfiler was not better than Cellpose.

Figure 7 The segmentation results of Cellpose and CellProfiler.

Why Field of View 1 is chosen in Figure 1 1f and View 2 chosen for fig 1g even though View 1 is larger than View 2?

We appreciate the inquiry. **Fig. 1g** was intended to allow the readers to visualize the results of cell segmentation, so field of View 1 was set as large as possible when the cell boundaries can be clearly seen. **Fig. 1f** was to allow the readers to visualize that SpeDe-Sr was able to remove noise better and enhance the details more reasonably. Focusing on a smaller field of view was more likely to make out the magnified details. If the field of view was as large as View 1, the readers would not be able to see the details of each cell as well as in View 2.

The authors stated that “All methods except SpiDe-Sr treated the CD8 (red pixel points) that was artificially judged as not being expressed in View 2 as effective information”. What is meant by artificially judged here?

We appreciate the inquiry. There were no cells in the black part of the View 2 and thus there shouldn't be any protein expressed on this part. However, we can see that there were red pixels in the black area in **Fig. 1f**, and we determined that these red pixels were background noise. Some of red aggregated areas were the cells that expressed CD8. So the expertise researcher judged that CD8 was not expressed in the region of View 2.

We are very grateful to the reviewers for the inquiry and we have revised the wording in the manuscript.

Page 5, line 149: All methods, except SpiDe-Sr, accidentally treated CD8 (red pixel points) that should not be expressed in View 2 as effective information.

The authors should visualize the learnt mappings by the networks for interpretability and explanation of the results.

We truly appreciate the reviewer's comment and suggestion. In fact, we had tried to visualize the middle layer features of the model, but this did not lead to a better understanding of the model by

the reader. The denoising module had 20 layers and 48 features were incorporated into the training. Four features output from the each of 9 middle layers were shown in **Figure 8**. It can be seen that the cellular features were effectively captured and maintained as the convolutional network deepens. For the SR module, which was a complex model combining 3 networks, the complex network structure made it very difficult to output meaningful variations of features layer by layer.

Figure 8 Four features output from each of the 9 middle layers of denoising module.

We put the output of the denoising module after different training times in Supplementary **Fig. 1d**, and we could see that the noise of the output image was gradually reduced. In Supplementary **Fig. 2d**, we put the outputs of the SR model after adjusting the blur kernel in each iteration, and we could see that the output was more and more clear. The end-to-end output with different number of blur kernel corrections may be more conducive to understanding the SR model.

According to the reviewer's inquiry, we have revised the Figure 8 to the Supplementary Information as Supplementary **Fig. 11** at Page 34, line 547.

Line #207, raw images were downsampled to 1/4th of original size. Why was this ratio chosen? Were other ratios tried by the authors? Preliminary results on the same can be included.

Thanks for the detailed questions. The reason for down-sampling the raw images to 1/4th of its original size was that our super-resolution module enlarges the images by a factor of four. In current super-resolution research, images were generally enlarged by a factor of 2, 4, or 8. For natural images, it was difficult to ensure that the image details were reasonable when enlarged 8 times using the model trained on real high-resolution images. For IMC images that required higher reasonableness and did not have real high-resolution images, it was difficult to get good results with the 8x magnification. In fact, we tried to train the super-resolution model by down-sampling the raw images to 1/8th of the original size, but the test results were not as desired. We then tried enlarging the images by a factor of 4 and were able to get results with reasonable details. As shown in **Figure 9**, we can see less artifacts and clearer cells in the 4x images compared to the 8x image.

In addition to this, in the real experiments, the size of the IMC image after 4x enlargement was similar to the size of the confocal image with the common magnification (20x, 0.4NA, resolution: 830nm).

Figure 9. Image enlargement of 8x (a)/ 4x (b).

There is no discussion on the statistical significance of the improved results.

Thanks for the comment. We have added the discussion of statistical significance to the manuscript.

Page5, line 146: The improvement of the PSNR and SSIM of the images, and cell extraction accuracy were statistically significant (paired-samples *t*-test, $P < 0.001$).

Page7, line 206: In e-h, asterisks indicated statistical significance by paired-samples *t*-test, $**P < 0.01$, $***P < 0.001$.

Page10, line 301: In **e-h**, asterisks indicated statistical significance by paired-samples *t*-test, ****P*< 0.001.

Page13, line 396: In **e-h**, asterisks indicated statistical significance by paired-samples *t*-test, **P*< 0.05, ***P*< 0.01, ****P*< 0.001

Page16, line 497: Red and blue asterisks respectively represent the statistical significance of proteins positively and negatively associated with LPS/ LTA versus LPS/ LTA (*t*-test, ***P*< 0.01, ****P*< 0.001).

Page28, line 966: The asterisk in the violin plots indicated the statistically significant difference between the two arrays as determined by two sided paired-samples *t*-test analysis. The *t*-tests were done in IBM SPSS Statistics 25 following standard procedure.

Page28, line 971: In **Fig. 5k**, Pearson correlation coefficients greater than 0.75 were marked with two asterisks and greater than 0.5 were marked with one asterisk. The number of replicates in each experiment was labeled in the figure legends.

Besides technical ambiguities, I have concerns about the biological data presented in the manuscript. For example, would authors explain why leukocyte marker CD45 can be detected in MCF7 breast cancer cells?

We appreciate the reviewer for the detailed comment. To address your question, we have checked the official website of ABcam for the antibodies we used (ab40763), and their positive control experiments included the MCF-7 cell line. The link to the website is <https://www.abcam.com/products/primary-antibodies/cd45-antibody-ep322y-ab40763.html>.

The full name of the protein corresponding to CD45 is Protein tyrosine phosphatase receptor type C (PTPRC). We looked up PTPRC on the Protein Atlas website and found that it was indeed expressed in the MCF-7 cell line. The result was shown in **Figure 10**. The link to the lookup is <https://www.proteinatlas.org/ENSG00000081237-PTPRC/cell+line>. It is an official website of the Protein Atlas program. In fact, according to the results found on the website, CD45 is expressed in a variety of breast cancer cell lines, such as CAL-51 and HDQ-P1. In addition, CD45 is also expressed in other tumor cell lines, such as brain cancer cell lines (DK-MG, SW1783), colorectal cancer cell lines (COLO320, COLO320DM).

Figure 10 Breast cancer cell lines expressing CD45.

Partial information of CD45 in the human protein atlas is shown in **Figure 11**.

Figure 11 The information of CD45 in the human protein atlas.

According to **Figure 11**, PTPRC (CD45) is a cancer-related gene, and is not only enriched in leukemia, but also in lymphoma. PTPRC is part of the Lymphoma-Humoral immune response cluster. It is probably because CD45 is a cancer-related gene and is expressed in a variety of tumor tissues so that it can be detected in MCF-7.

In addition, we also repeated the same experiment as in the manuscript again. The raw IMC and confocal images were shown in **Figure 12**. The results of our experiment confirmed that CD45 was expressed in MCF-7.

a. Confocal images

b. IMC image

Figure 12 a, Immunofluorescence images of CD45 in MCF-7. **b**, IMC image of MCF-7 labeled CD45. Green (nucleus)/ Red (CD 45).

In addition, we also did Westen Blot for CD45 on MCF-7, the result was shown in **Figure 13**. The Westen Blot result also indicated that CD45 was expressed in MCF7. The raw images of Westen Blot were uploaded as an additional supplementary file.

Figure 13. Westen Blot for CD45 on MCF-7.

And we sent the MCF-7 cell line from our laboratory to a biotechnology company for STR (Short Tandem Repeat) cell line identification. The identification report showed that our cell line was indeed MCF-7 cell line. The STR identification report was uploaded as an additional supplementary file.

Immune cells and epithelial cells are morphologically very different. Could the authors, use two cell types with more similar morphology as the basis of their experiments. Lets say CD8 vs CD19?

We appreciate the reviewer for the comment. Our understanding of this comment is that we need to experiment again with other cell lines that have similar morphology to MCF-7. In addition to this, the effect of the model on images with both CD8 and CD19 proteins needs to be seen. Accordingly, we did the dual-labeling experiment again by using the LINCAP cells that were most morphologically similar to MCF-7 among the cell lines available in our lab. The results were shown **Figure 14**. We can see that the noise caused by the nonspecific adsorption of the antibody was removed after SpiDe-Sr, and the nuclei and borders of the cells were clearer. The yellow arrows pointed to the noise created by nonspecific adsorption rather than cells. In the raw image, the noise was detected as cell, and there were more missing cells in the yellow box.

Figure 14 The fluorescent/ metal-dual labeling experiment with LNCAP cells. The right side images were the cell segmentation results of the left side images.

We labeled CD8 and CD19 on the same sample, and the images before and after SpiDe-Sr were shown in **Figure 15**. We can see that the background noise and the noise caused by the nonspecific adsorption of the antibodies were removed, and the boundaries and the nuclei of cells were clearer after SpiDe-Sr.

Figure 15. IMC images before and after SpiDe-Sr on the same breast cancer sample labeled with CD8 and CD19.

Overall, we would like to express our gratitude again to both reviewers for their time in reading and facilitating our manuscript. We sincerely appreciate the insightful comments and suggestions. We hope our additional experimental data and elaborations may adequately address the reviewer's questions and inquiries.

Reviewers' Comments:

Reviewer #1:

Remarks to the Author:

I carefully read the comments of other reviewers, the revised paper, and the author's responses. In fact, I didn't raise too many questions. I just hope that the author can further clarify and explain some issues. The author adds some supplementary explanations. From my perspective, I think technically this article holds up. The author gives some work to demonstrate the significance of this task. Given that there is already so much relevant literature, it seems that this task makes sense. I'm still not quite sure though.

Reviewer #2:

Remarks to the Author:

The authors adequately responded to all the reviewer comments. I would recommend the manuscript for publication. At the same time, I would suggest an extensive editorial revision of the writing.